# Catalytic *4-exo-dig* carbocyclization for the construction of furan-fused cyclobutanones and synthetic applications

Kemiao Hong[1,3], Yi Zhou[2,3], Haoxuan Yuan[1], Zhijing Zhang[1], Jingjing Huang[1], Shanliang Dong[1], Wenhao Hu [1], Zhi-Xiang Yu [2] ✉ & Xinfang Xu [1] ✉

Cyclobutanone is a strained motif with broad applications, while direct assembly of the aromatic ring fused cyclobutanones beyond benzocyclobutenone (BCB) skeletons remains challenging. Herein, we report a Rh-catalyzed formal [3+2] annulation of diazo group tethered alkynes involving a *4-exo-dig* carbocyclization process, providing a straightforward access to furan-fused cyclobutanones. DFT calculations disclose that, by comparison to the competitive *5-endo-dig* process, 4-exo-dig carbocyclization is mainly due to lower angle strain of the key *sp*-hybridized vinyl cationic transition state in the cyclization step. Using less reactive catalysts $Rh_2(carboxylate)_4$ is critical for high selectivity, which is explained as catalyst-substrate hydrogen bonding interaction. This method is proved successful to direct access previously inaccessible and unknown furan-fused cyclobutanone scaffolds, which can participate in a variety of post-functionalization reactions as versatile synthetic blocks. In addition, preliminary antitumor activity study of these products indicates that some molecules exhibit significant anticancer potency against different human cancer cell lines.

Cyclobutanone is a strained cyclic motif[1–4] that is present in many natural products[5–8], bioactive molecules[9–11], and materials[12,13]. By virtue of its high tendency to release strain energy, four-membered cyclic ketone derivatives have made possible a myriad of synthetic methods through different catalytic ring-opening transformations[1–4,14–16]. Among these advances, benzocyclobutenones (BCBs)[17,18], which possess higher ring strain than saturated cyclobutanones due to their rigid planar architectures, are versatile building blocks for the transition-metal-mediated C-C bond activation to access diverse skeletons that are nontrivial to synthesize otherwise[19,20]. Although many efforts have been devoted to the synthesis of BCB skeletons in the past few decades, including [2+2]-cycloadditions[21,22], coupling reactions[23,24], and few others[25,26] (Fig. 1a, path a–d), the construction of cyclobutanones that are fused to nonbenzene aromatics remains an uncharted area in synthetic chemistry. For example, the furan-fused cyclobutanone **A** (Fig. 1a), which might provide an intriguing opportunity for the discovery of untapped properties and reactivities, is an unknown molecule and has not yet been prepared. Therefore, the development of practical methods for the directly assembly of these unique fused architectures with structural diversity from readily available materials has long been a demanding synthetic challenge.

In the past two decades transition-metal complex catalyzed alkyne carbocyclization has emerged as a powerful method in organic synthesis for the direct construction of carbocyclic and heterocyclic structures from different functionalized alkyne precursors[27–36]. The carbene/alkyne metathesis (CAM)[37,38] cascade reaction is one of the elegant cyclization protocols for the straightforward preparation of 5- and 6-membered carbocyclic structures through different types of

[1]Guangdong Key Laboratory of Chiral Molecule and Drug Discovery, School of Pharmaceutical Sciences, Sun Yat-sen University, Guangzhou 510006 Guangdong, China. [2]Beijing National Laboratory for Molecular Sciences (BNLMS), Key Laboratory of Bioorganic Chemistry and Molecular Engineering of Ministry of Education, College of Chemistry, Peking University, 100871 Beijing, China. [3]These authors contributed equally: Kemiao Hong, Yi Zhou. ✉e-mail: yuzx@pku.edu.cn; xuxinfang@mail.sysu.edu.cn

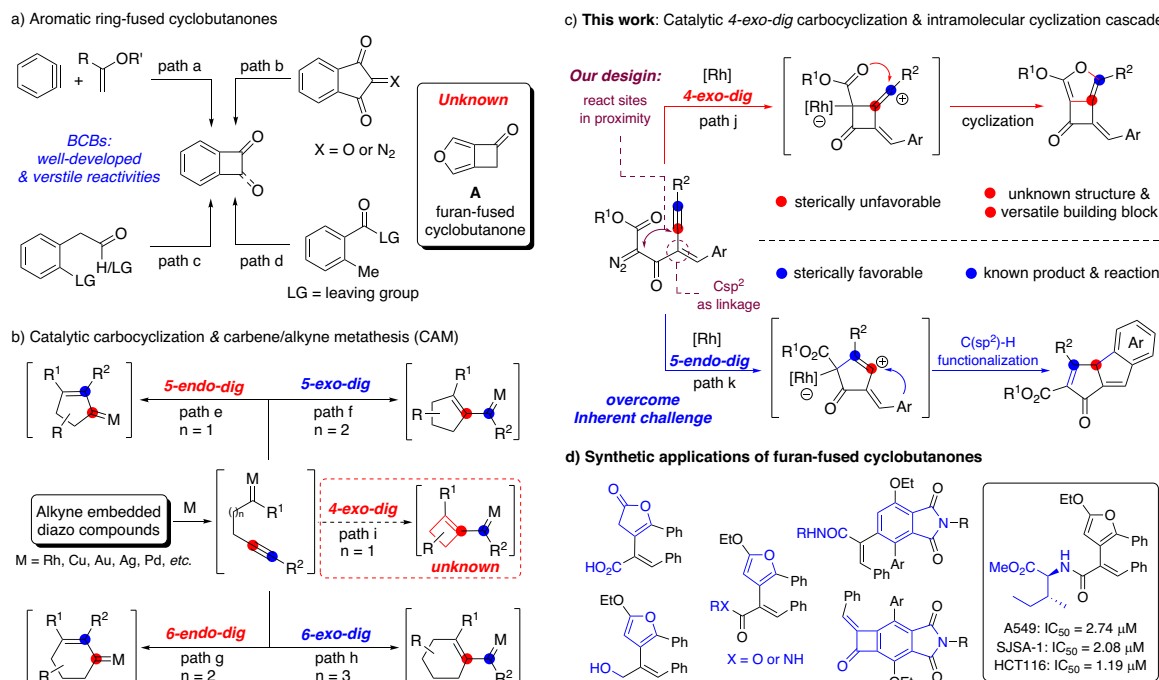

**Fig. 1 | Catalytic cycloaddition & cyclization methods. a** Construction of aromatic ring-fused cyclobutanones & unknown structure of furan-fused cyclobutanone. **b** Catalytic carbocyclization & carbene/alkyne metathesis (CAM). **c** This study: catalytic *4-exo-dig* carbocyclization process for the direct construction of furan-fused cyclobutanones *vs* competitive *5-endo-dig* cyclization. **d** Synthetic applications of synthesized furan-fused cyclobutanones.

*exo-* or *endo-dig* cyclizations (Fig. 1b, path e−h)[39–52]. However, only limited examples have been disclosed for the formation of small ring systems[53–55], despite its potent capability in synthetic applications. The *4-exo-dig* cyclization process is certainly the least-explored one, and no successful example has been disclosed thus far (Fig. 1b, path i), which might be due to the competitive *5-endo-dig* pathway that was proposed to be more sterically favorable.

Inspired by these encouraging works and continuing our interests in catalytic alkyne functionalization[38,56–58], we were curious about the feasibility of enabling an uncharted catalytic *4-exo-dig* cyclization process for the direct construction of four-membered structures. To the best of our knowledge, the catalytic *4-exo-dig* carbocyclization of alkynes for the construction of cyclobutanones has not been reported, although rare examples via radical type cyclization[53] or coupling reactions[54,55] have been disclosed for the synthesis of four-membered analogous.

Herein, we present our recent results in this direction, a Rh-catalyzed formal [3+2] annulation of diazo group tethered alkynes involving a *4-exo-dig* carbocyclization process. This methodology provides direct and convenient access to the synthesis of furo[3,4-*b*] cyclobutanones via an intramolecular annulation of the in-situ formed zwitterionic intermediates rather than through a CAM process involving vinyl carbene species (Fig. 1c, path j). In our protocol, a vinylic sp²-hybridized carbon has been introduced as the key linkage of the diazo group and the triple bond, instead of using a flexible aliphatic chain, which could facilitate the *4-exo-dig* carbocyclization by bringing the reacting centers in close proximity. Thus, one of the serious drawbacks, the inherent propensity of the competitive *5-endo-dig* carbocyclization of alkynes towards thermodynamically stable tricyclic products through the endocyclic carbene intermediates, could be overcome (path k). Moreover, density functional theory (DFT) computation elucidates that the favorable *4-exo-dig* cyclization is mainly attributed to the relatively lower angle strain of the key sp-hybridized vinyl cationic transition state in the cyclization step. The synthesized cyclobutanone-fused furan derivatives with imbedded furyl, carbonyl, and benzylidene motifs have not been previously reported and are

potential versatile synthetic building blocks[17,18]. This practical method could be successfully applied for the late-stage modification of complex natural products and pharmaceutical molecules, and the synthetic application of these unique compounds has been demonstrated through a variety of ring-opening reactions and cycloadditions for the diversity-oriented-synthesis (Fig. 1d).

## Results and discussion
### Optimization of the reaction conditions
We began our investigation by pursuing the optimal reaction conditions for the catalytic *4-exo-dig* carbocyclization process with enynone **1a**, tethered with a diazo moiety by using a vinylic sp²-hybridized carbon linkage, as the model substrate (Table 1). Various metal catalysts were initially examined in dichloromethane (DCM) at 40 °C. Zinc chloride, $ZnCl_2$, was found catalytically inactive in this reaction (entry 1), whereas Ag(I) and Au(I) catalysts led to the decomposition of the material into a complex mixture (entries 2 and 3). When Cu(I) and Pd(II) complexes were introduced, the *4-exo-dig* cyclized product **2a** was formed in 15% and 21% yields, respectively (entries 4 and 5). The yields could be further improved by using the dirhodium complex, $Rh_2(OAc)_4$, as the catalyst, affording the furan-fused cyclobutanone **2a** in 44% yield contaminated with tricyclic product **3a** in 20% yield (entry 6). Encouraged by these results, a variety of dirhodium complexes were screened (entries 7–14), and the superior results were obtained with $Rh_2(OPiv)_4$, generating **2a** in 54% yield (entry 9). Then, different solvents were investigated with $Rh_2(OPiv)_4$ as the metal catalyst (entries 15–18), showing that the yields could be enhanced to 81% by conducting the reaction in ethyl acetate (entry 18, EtOAc). Comparable high yields could be obtained with $Rh_2(OAc)_4$ as the catalyst under otherwise identical conditions (entry 19, 80% yield). Furthermore, the catalyst loading could be reduced to 1.0 mol% without any deleterious effect on the reaction yield and selectivity (entry 20). Using 4 Å MS as additive and $Rh_2(OPiv)_4$ as the catalyst in EtOAc, the best results, in terms of yield and selectivity, were obtained for the formation of **2a** (entry 21, 88% NMR yield and 84% isolated yield). In addition, no reaction occurred in the absence of dirhodium catalyst, which

## Table 1 | Optimization of the reaction conditions

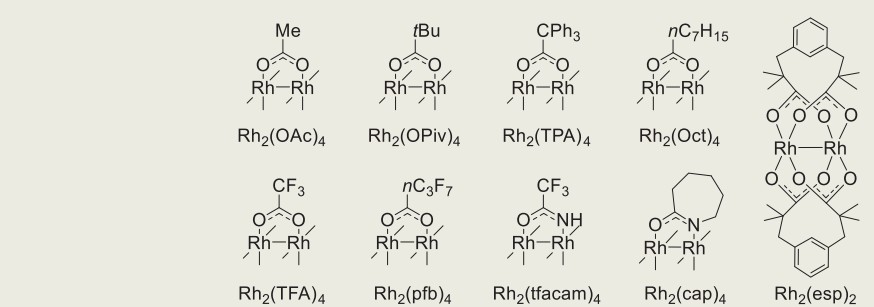

| Entry | Catalyst (x mol%) | Solvent | Yield (%)[a] | |
|---|---|---|---|---|
| | | | **2a** | **3a** |
| 1[b] | ZnCl₂ (20.0) | DCM | <5 | 6 |
| 2[c] | AgSbF₆ (5.0) | DCM | – | – |
| 3[c] | JohnPhos(MeCN)AuSbF₆ (5.0) | DCM | – | – |
| 4 | Cu(MeCN)₄PF₆ (5.0) | DCM | 15 | 30 |
| 5 | [PdCl(η³-C₃H₅)]₂ (5.0) | DCM | 21 | <5 |
| 6 | Rh₂(OAc)₄ (2.0) | DCM | 44 | 20 |
| 7 | Rh₂(Oct)₄ (2.0) | DCM | 45 | 19 |
| 8 | Rh₂(TPA)₄ (2.0) | DCM | 41 | 15 |
| 9 | Rh₂(OPiv)₄ (2.0) | DCM | 54 | 10 |
| 10 | Rh₂(esp)₂ (2.0) | DCM | 43 | 9 |
| 11 | Rh₂(pfb)₄ (2.0) | DCM | 20 | 17 |
| 12 | Rh₂(TFA)₄ (2.0) | DCM | 16 | 14 |
| 13 | Rh₂(tfacam)₄ (2.0) | DCM | 20 | 65 |
| 14 | Rh₂(cap)₄ (2.0) | DCM | NR | NR |
| 15 | Rh₂(OPiv)₄ (2.0) | DCE | 74 | 13 |
| 16 | Rh₂(OPiv)₄ (2.0) | toluene | 47 | 12 |
| 17 | Rh₂(OPiv)₄ (2.0) | CH₃CN | 71 | 10 |
| 18 | Rh₂(OPiv)₄ (2.0) | EtOAc | 81 | <5 |
| 19 | Rh₂(OAc)₄ (2.0) | EtOAc | 80 | <5 |
| 20 | Rh₂(OPiv)₄ (1.0) | EtOAc | 83 | <5 |
| 21[d] | Rh₂(OPiv)₄ (1.0) | EtOAc | 88 (84)[e] | <5 |

Reaction conditions: to a solution of metal catalyst (x mol%) in 1.0 mL solvent, was added **1a** (0.1 mmol, 34.4 mg) in the same solvent (1.0 mL) slowly over 10 min via a syringe at 40 °C under an argon atmosphere, and the reaction mixture was stirred for additional 1.0 h under these conditions.

*DCM* dichloromethane, *DCE* dichloroethane, *EtOAc* ethyl acetate.

[a]Yields were detected by proton NMR of the crude reaction mixture using 4-nitrobenzaldehyde as the internal standard.

[b]Low conversions (<20%) were observed, and most of the material was recovered.

[c]The material **1a** was decomposed into a complex mixture.

[d]100 mg 4 Å MS was added.

[e]Isolated yield.

indicated that the reaction initiated via the formation of metal carbene species.

### Scope of the *4-exo-dig* carbocyclization reaction

With the optimized reaction conditions in hand, we next examined the generality of this formal [3+2] annulation reaction with different alkyne-tethered diazo compounds. As shown in Fig. 2, different substitutions on the aryl ring attached to the alkyne had little effect on the reaction outcomes, leading to a range of furan-fused

cyclobutanones bearing electron-neutral, -rich, or -deficient substituents on the different positions of the aromatic ring in high to excellent yields (**2a–2p**, 60–92%). Likewise, the 1-naphthyl, 1-thienyl, ferrocenyl, and cyclohexenyl groups substituted enynones underwent the reaction smoothly, delivering products **2q–2t** in 74–85% yields. Notably, alkyl halide is also a compatible substituent under these conditions, forming the chlorinated product **2u** in 64% yield. The substitution pattern on the styryl moiety was then examined, and the corresponding products were obtained in

**Fig. 2 | Substrate scope for the synthesis of furan-fused cyclobutanones 2.**
Reaction conditions: to a solution of Rh$_2$(OPiv)$_4$ (1.2 mg, 1.0 mol%), and 4 Å MS (100 mg) in EtOAc (1.0 mL), was added a solution of diazo compounds **1** (0.2 mmol) in EtOAc (1.0 mL) slowly over 10 min via a syringe under an argon atmosphere at 40 °C, and the reaction mixture was stirred for additional 1.0 h under these conditions. The yields are isolated yields. [a]The reaction was carried out on a 5.0 mmol scale with 0.25 mol% of Rh$_2$(OPiv)$_4$ loading. [b]The reaction was carried out in the presence of Rh$_2$(esp)$_2$ (1.0 mol%) in DCE at 80 °C.

generally high yields (**2v**–**2x**, 83–85%) regardless of electronic or steric effects. The variation of the ester part had a negligible influence on the reactivity, giving the *tert*-butyl, benzyl, homopropargyl, and terminal alkenyl derivatives **2y**–**2ab** in high yields. Furthermore, *D*-menthol and estrone derived substrates also worked very well under the optimal conditions, generating the corresponding products **2ac** and **2ad** 75% and 80% yields, respectively. Terminal alkyne and TMS substituted one did not give the desired product under current conditions, although all the materials were consumed. Notably, the pyrrole-fused cyclobutanone **2ae** was formed in 42% yield when [1,2,3]triazolo[1,5-*a*]quinoline derivative was used as the carbene precursor in the presence of Rh$_2$(esp)$_2$ (1.0 mol%) in DCE at 80 °C (see note b in Fig. 2 for details). We could also replace the styryl moiety with the cyclopropane unit, delivering the product **2af** in 91% yield. Rather than the ester substrates, the ketone analogous performed well under current conditions, forming the desired product **2ag** in 72% isolated yield. The amide derivative did not form the desired cyclized product, but gave the eneyne products **2ah** and **2ah'** in combined high yield, which formed *via* a Wolff rearrangement, nucleophilic addition with H$_2$O,

decarbonation, and isomerization sequence. The structures of products **2g** and **2af** were confirmed by single-crystal X-ray diffraction analysis. In addition, this reaction could be conducted on a gram scale (5.0 mmol), providing 1.27 g **2a** in 80% yield (note a).

## Synthetic applications

The furan-fused cyclobutanone products obtained from this reaction were expected to serve as precursors to other versatile and useful synthetic building blocks owing to the rich ring-opening chemistry of this unique structure on the carbonyl group[18,59]. Thus, we performed a variety of ring-opening transformations of the strained cyclobutanone unit in the product **2a** to demonstrate the utility of this method (Fig. 3). For example, addition of **2a** with hydroxide ion followed by treatment with HCl gave lactone product **4** in 81% yields. Next, reduction with NaBH$_4$, or addition with Grignard reagent, formed the primary and tertiary alcohols **5a** and **5b** in 94% and 90% yields, respectively. Furthermore, addition with alcohols directly provided ester products **6a** and **6b** in high yields. In addition, when trimethylsulfoxonium iodide was used as carbonic nucleophile, the ring-opening product **7** was generated in 50% yield in the presence of NaH; when diazoamide was

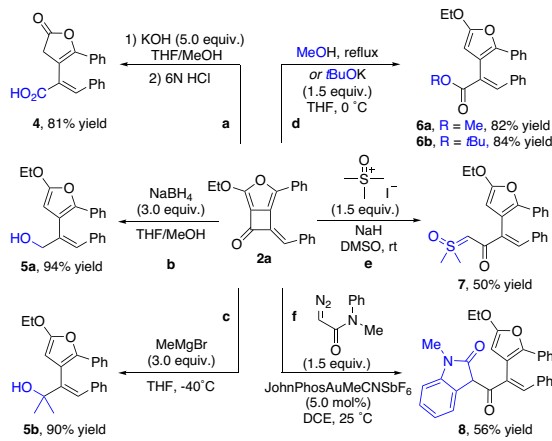

**Fig. 3 | Synthetic applications of furan-fused cyclobutanone 2a. a** Synthesis of **4**. **b** Synthesis of **5a**. **c** Synthesis of **5b**. **d** Synthesis of **6a** and **6b**. **e** Synthesis of **7**. **f** Synthesis of **8**.

used as the reagent, which forms the zwitterionic intermediate in situ in the presence of gold catalyst, the ring-opening product **8** was formed in 56% yield.

Encouraged by above results, we envisioned that these highly reactive fused cyclobutanone derivatives might be capable of late-stage functionalization (LSF) for different nucleophilic substrates under mild reaction conditions. After brief optimization, an extremely wide range of commercially available amines and alcohols, including those derived from natural products, underwent ring-opening addition to give the amide and ester products[60], respectively (Fig. 4). Beyond the regular inert alkyl primary and secondary amines, which led to the amide products **9–12**, in 71–92% yields, the mild conditions tolerated a wide range of functional groups, including terminal alkyne (**13**), ether (**14**), acetal (**15**), pyridyl (**16**), indolyl (**17**), and cyan (**18**) groups, forming the corresponding amides in high to excellent yields. Notably, a variety of chiral amino alcohols could be used as nucleophiles, selectively resulting the ring-opened amide products that contained a free hydroxyl group (**19–23**, in >82% yields). Moreover, a number of natural and synthetic amino acid derived esters were all suitable substrates, producing the furan-containing amino acid derivatives in good to high yields (**24–49**). In the case with *L*-ornithine methyl ester, an ester-amide exchange occurred with the ring-opening amide product, leading to valerolactam **44** in 65% yields. The broad scope of this mild ring-opening transformation inspired us to explore the feasibility of the current strategy to the late-stage modification of complex natural products and pharmaceutical molecules. To our delight, tropine, stigmasterol, and cholesterol all proceeded well under modified basic conditions, producing the corresponding ester derivatives (**50–52**) in >85% yields; meanwhile, the ring-opening reactions with bioactive molecules containing an amino group could be smoothly applied to the current conditions, generating the amide products in 68–95% yields (**53–58**). The structure of **38** was ambiguously assigned by X-ray crystallography. The successful application of the drugs and sterols mentioned above in this ring-opening reaction demonstrated the potent synthetic value of this method, which might be used as a potential click reaction in bioorthogonal chemistry.

Beyond the cyclobutanone unit, the presence of different combinations of functional groups renders these fused structures to be versatile synthetic building blocks. The furan motif, which is an electron-rich aromatic ring, could undergo a Diels-Alder reaction with electron-deficient alkenes followed by aromatization, providing a convenient access to poly-substituted benzocyclobutenones (BCBs), which would be difficult to access otherwise[17,18,21–26]. After a brief optimization, the tricyclic BCB derivatives **59–62** were obtained in

61–66% yields via an intermolecular Diels-Alder reaction of synthesized cyclobutanones **2** with *N*-substituted maleimides in a mixed solvent (toluene/DCE = 10/1) at 90 °C for 12 h (Fig. 5). The structure of product **59** was confirmed by X-ray crystallography. Furthermore, these obtained BCBs could also be subjected to the ring-opening transformation. For example, treatment of these products with piperidine in DCM at 60 °C, generated cinnamamide derivatives **63–69** in synthetically useful yields (Fig. 5).

## Anticancer activity evaluation

Moreover, we have selected a few of synthesized compounds (**6a, 9–12, 14, 15, 18–21, 23–40, 42, 43, 46, 47, 49–55, 63–66**, and **68**) for the anticancer activity evaluation on cell viability via the CCK8 assay for HCT116 (colon cancer), MCF-7 (breast cancer), A549 (lung adenocarcinoma), SJSA-1 (osteosarcoma cancer), and MDA-MB-231(breast cancer) human cancer cell lines (see Supplementary Tables 1 and 2 in the Supplementary Information for details). The results show that compounds **21** (HCT116 cells, $IC_{50} = 2.33 \mu M$; MCF-7 cells, $IC_{50} = 6.85 \mu M$; SJSA-1 cells, $IC_{50} = 3.47 \mu M$), **29** (HCT116 cells, $IC_{50} = 1.19 \mu M$; A549 cells, $IC_{50} = 2.74 \mu M$; SJSA-1 cells, $IC_{50} = 2.08 \mu M$), and **50** (HCT116 cells, $IC_{50} = 4.42 \mu M$; MCF-7 cells, $IC_{50} = 4.23 \mu M$; A549 cells, $IC_{50} = 6.43 \mu M$; SJSA-1 cells, $IC_{50} = 4.30 \mu M$; MDA-MB-231 cells, $IC_{50} = 6.88 \mu M$) exhibited significant and broad anticancer potency (Fig. 6). A further structure–activity relationship study with various human cancer cell lines is ongoing in the laboratory.

## Mechanistic investigations

Control experiments were conducted to investigate the reaction mechanism of this intriguing process (Fig. 7). Initially, the reaction of **1af** in the presence of excess amount of styrene under standard conditions was carried out, which intended to obtain the cyclopropanation product with possible carbene intermediate(s). However, only furan-fused cyclobutanone **2af** was formed in 71% yield, and no cyclopropane product was observed by mass spectrometry (MS) analysis of the crude reaction mixture (Fig. 7a), which implies that the intramolecular cascade reaction might be more favorable even if the second carbene species was formed via the CAM process. On the contrary, the corresponding addition product with BnOH was isolated in 31% yield combined with **2af** in 40% yield (Fig. 7b), and this cyclobutanone product **70** was confirmed by X-ray crystallography after converting to the ring-opening product **71** (CCDC 2249373, 2246883, 2110463, 2083438, and 2265899 contain the supplementary crystallographic data for **2g**, **2v**, **38**, **59**, and **71**. These data can be obtained free of charge from the Cambridge crystallographic data centre *via* www.ccdc.cam.ac.uk/data_request/cif) (Fig. 7c). These results suggested potential carbene or cationic intermediates in this cascade reaction, and the density functional theory (DFT) calculations support the cationic intermediate (see Supplementary Fig. 236 in SI for details).

The catalyst effects on the regioselectivity of this reaction was studied through DFT calculations (Supplementary Data 1). To understand the intrinsic selectivity of this reaction, we computed the two competing pathways to **2a** and **3a** using $Rh_2(HCOO)_4$ as the model catalyst and **1a** as the substrate, and all the structures were obtained in the gas phase. We proposed the reaction mechanisms (shown in Fig. 8), which have considered our calculation results and previous studies[37–51]. Initially, the acceptor/acceptor rhodium carbene intermediate **Int1** is generated by extrusion of nitrogen from the diazo compound **1a** in the presence of dirhodium complex (the overall activation free energy of this step is 16.5 kcal/mol, which should be the rate-determined step of this reaction, see Supplementary Fig. 232 in SI for details). Then **Int1** undergoes electrophilic 4-*exo-dig* cyclization via **TS1** to afford the vinyl cationic species **Int2**, requiring an activation free energy of 0.6 kcal/mol. After that, **Int2** undergoes an intramolecular cyclization via **TS2** to afford product **2a** with an activation free energy of 3.8 kcal/mol. Another pathway to product **2a** via carbene/alkyne metathesis (via the

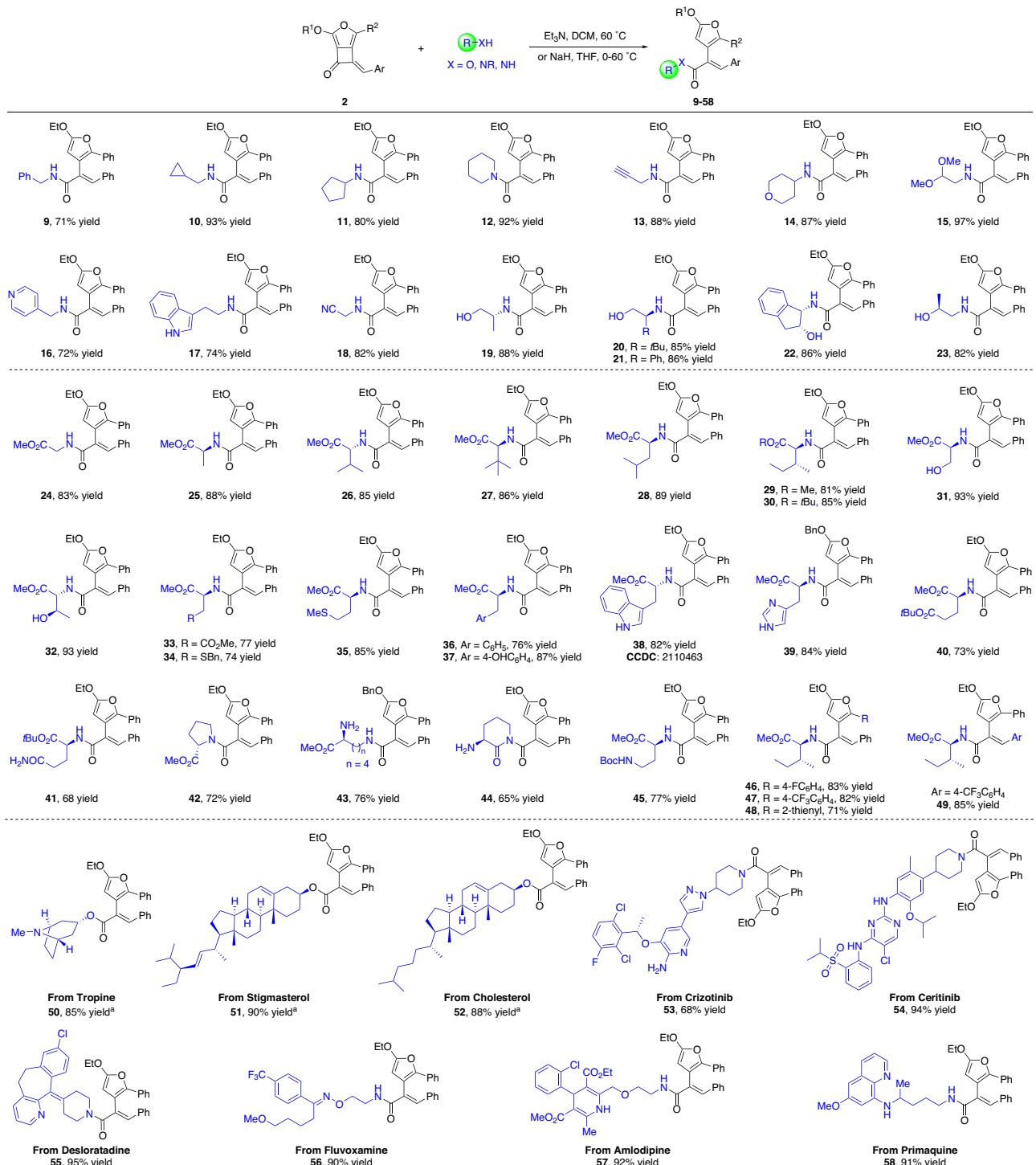

**Fig. 4 | Synthesis of furan-embedded amide and ester derivatives.** Reaction conditions: **2** (0.1 mmol), amines or hydrochloride of amino acid esters (0.15 mmol, 1.5 equiv.), and Et₃N (0.2 mmol, 2.0 equiv.) in DCM (2.0 mL), and stirring for 48 h in an oil bath at 60 °C, unless otherwise noted. ᵃ**2** (0.1 mmol), alcohols (0.15 mmol, 1.5 equiv.), and NaH (0.15 mmol, 1.5 equiv.) in dry THF (2.0 mL) at 0 °C, then warmed to 60 °C slowly and stirred for 1.0 h. All yields are isolated yields.

1,3-Rh migration through a singlet transition state **TS3** or intersystem crossing (ISC) process is disfavored, because **TS3** is higher in energy than **TS2** by 4.5 kcal/mol (see Supplementary Figs. 233 and 234 in SI for the full energy surface)[44]. The competing pathway leading to **3a** starts from 5-*endo-dig* cyclization via **TS4**, which requires an activation free energy of 5.0 kcal/mol and converts **Int1** into the vinyl cation **Int3**. Then, **Int3** gives the final 6-5-5 tricyclic product **3a** through a 1,3-Rh migration and an aromatic C-H insertion sequence via an endocyclic rhodium-carbene intermediate **Int4** (See Supplementary Information

for the full energy surface). As shown in Fig. 8, the first electrophilic cyclization steps via **TS1** or **TS4** determine the regioselectivity of this reaction, and the formation of **2a** is favored over the formation of **3a** because **TS1** is lower in energy than **TS4** by 4.4 kcal/mol.

Then the catalyst effects on the regiochemistry were investigated by computing the activation energies of carbene complexes generated from **1a** with catalysts of Rh₂(HCOO)₄, Rh₂(OAc)₄, and Rh₂(tfacam)₄, respectively (Fig. 9). Rh₂(OAc)₄ was chosen because of its comparable regioselectivity in EtOAc (**2a:3a** > 16:1) with the used Rh₂(OPiv)₄

(**2a:3a** > 17.6:1), and much less conformers compared to those from Rh$_2$(OPiv)$_4$, which has 12 rotated methyl group. Catalyst Rh$_2$(tfacam)$_4$ has different ligands and affords inverse selectivity (**2a:3a** = 1:3.6 in EtOAc), prompting us to investigate the reasons behind this phenomenon. Considering that solvation effects play an important role in affecting the regiochemistry (The axial effect has also been considered but showed insignificant effect on the regioselectivity of this reaction, see Supplementary Fig. 235 in SI for details)[61], all the structures were obtained in solution rather than in the gas phase. The DFT calculations predict that, when Rh$_2$(HCOO)$_4$ is used as catalyst, 4-*exo-dig* cyclization (via **TS1** requiring an activation free energy of 2.8 kcal/mol) is favored over the 5-*endo-dig* cyclization (via **TS4** requiring 4.2 kcal/mol) by 1.4 kcal/mol, giving **2a** as the major product. While using the real catalyst of Rh$_2$(OAc)$_4$, 4-*exo-dig* cyclization (via **TS1-Me** requiring an activation free energy of 4.4 kcal/mol) is favored over the 5-*endo-dig* cyclization (via **TS4-Me** requiring 5.6 kcal/mol) by 1.2 kcal/mol, indicating that **2a** is the major product and the computed ratio of **2a/3a** is 7.4:1. This is consistent with the experimental results, which gave the 4-membered ring structure **2a** as the major product. Catalyst Rh$_2$(OAc)$_4$ is less reactive than Rh$_2$(HCOO)$_4$ for both cyclization reac-

tions, which can be understood by considering the electron-donating effect of methyl groups in Rh$_2$(OAc)$_4$, making the carbene center less electrophilic (natural population analysis (NPA) charge of carbene center reduced from +0.063 in **Int1** to +0.053 in **Int1-Me**). Here, 5-*endo-dig* cyclization forming the five-membered ring for both Rh$_2$(HCOO)$_4$ and Rh$_2$(OAc)$_4$ as catalysts are disfavored by 1.2–1.4 kcal/mol with respect to the 4-*exo-dig* cyclization forming four-membered ring, which is counter-intuitive, considering that the latter forming a smaller ring is usually disfavored than the former forming a five-membered ring. We attribute this to the formation of a strained vinyl cation[62,63] in 5-*endo-dig* cyclization. The bond angles at the *sp*-hybridized vinyl cationic center in **TS4** and **TS4-Me** are around 146°, which are far from the required bond angle of 180°.

However, when the catalyst Rh$_2$(tfacam)$_4$ is used, DFT calculations show that the activation free energies of both cyclization processes are decreased to only 1.5–1.6 kcal/mol, while using Rh$_2$(HCOO)$_4$ and Rh$_2$(OAc)$_4$ as the catalysts, the activation free energies are 2.8–5.6 kcal/mol. The hydrogen bonding interactions between the catalyst's amide group and the substrate's carbonyl group (supported by non-covalent interactions (NCI) analysis[64]) stabilize the transition states of **TS1-CF3** and **TS4-CF3**, and lower activation free energies are then required. While for catalyst Rh$_2$(HCOO)$_4$ and Rh$_2$(OAc)$_4$, their carboxylate groups cannot form hydrogen bonding with substrate and higher activation free energies are needed. Therefore, the transition states for both 4-*exo-dig* and 5-*endo-dig* cyclizations using catalyst Rh$_2$(tfacam)$_4$ are earlier and have less energy difference (here is 0.1 kcal/mol based on the Hammond postulate[65]). This is consistent with experimental observation of low regioselectivity. Low active catalysts of Rh$_2$(HCOO)$_4$ and Rh$_2$(OAc)$_4$ then can have later transition states and higher selectivity for **2a** *vs* **3a**.

In summary, we have disclosed a dirhodium-catalyzed *4-exo-dig* carbocyclization/[3 + 2] annulation cascade reaction of alkyne-tethered diazo compounds, which provides a straightforward approach for the synthesis of structurally novel furan-fused cyclobutanone scaffolds that are nontrivial to construct otherwise. This reaction features a rare example of catalytic 4-*exo-dig* carbocyclization with high chemo- and regio-selectivity, in spite of many possible competitive pathways. Density functional theory (DFT) computation elucidated that the favorable 4-*exo-dig* carbocyclization is mainly attributed to the lower angle strain of the key *sp*-hybridized vinyl cationic transition state in the cyclization step in comparison to the competitive 5-*endo-dig* process. Calculations revealed that, compared to catalyst Rh$_2$(tfacam)$_4$, catalysts of Rh$_2$(carboxylate)$_4$ are less reactive/more

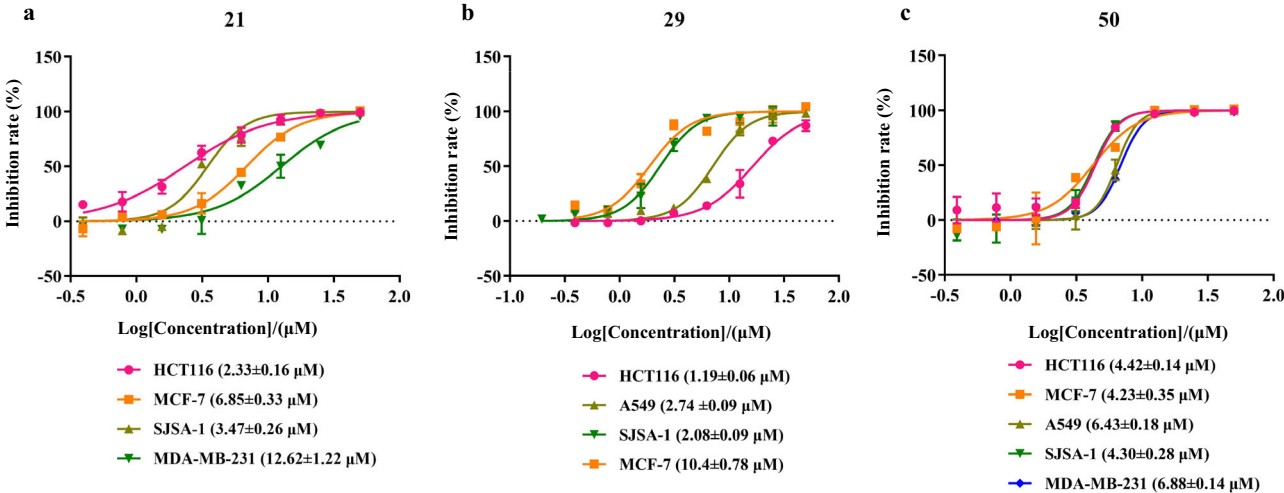

**Fig. 5 | Synthesis of benzocyclobutenones and poly-substituted benzene derivatives.** Reaction conditions: **2** (0.1 mmol), and maleimides (0.15 mmol, 1.5 equiv.) in a mixed solvent of toluene/DCE = 10/1 (1.0 mL) at 90 °C for 12 h. All yields are isolated yields. ᵃBenzocyclobutenones (0.1 mmol), and piperidine (0.15 mmol, 1.5 equiv.) in DCM (2.0 mL), and stirring for 24 h in an oil bath at 60 °C.

**Fig. 6 | The cytotoxicity of the synthetic compounds in various human cell lines. a** The IC$_{50}$ curves of compound **21** on the inhibition of various human tumor cells. **b** The IC$_{50}$ curves of compound **29** on the inhibition of various human tumor cells. **c** The IC$_{50}$ curves of compound **50** on the inhibition of various human tumor cells.

selective for the reaction. The reason is due to the facts that, Rh$_2$(tfacam)$_4$ with amide ligand can form hydrogen bonding with the substrate to stabilize the regio-determining transition states (so the reactions become more reactive and less selective), while catalysts of Rh$_2$(carboxylate)$_4$ with carboxylate ligands, don't have such stabilizing hydrogen bond interactions with the substrate (less reactive, more selective). The synthesized fused cyclobutanone derivatives, which are prepared with different combinations of functional groups, make possible a variety of ring-opening and cycloaddition reactions as versatile synthetic building blocks for diversity-oriented-synthesis. Further synthetic applications, including but not limited to novel transformations development and bioactive molecule discovery, could be expected in due course.

## Methods

### General procedure for the synthesis of furan-fused cyclobutanones 2

To a 10-mL oven-dried vial containing a magnetic stirring bar, Rh$_2$(OPiv)$_4$ (1.2 mg, 1.0 mol%), and 4 Å MS (100 mg) in EtOAc (1.0 mL), was added as a solution of diazo compounds **1** (0.2 mmol) in the EtOAc (1.0 mL) slowly via a syringe at 40 °C under argon atmosphere. After addition, the reaction mixture was stirred for additional 1.0 h under these conditions. Until consumption of the material (monitored by TLC), the reaction mixture was purified by column chromatography on silica gel without any additional treatment (Hexanes: EtOAc = 15:1 to 5:1) or recrystallized from MeOH to give the pure products **2**.

### General procedure for the synthesis of amide derivatives 9−49 and 53−58

To a 10-mL oven-dried vial containing a magnetic stirring bar, cyclobutanones **2** (0.1 mmol), amine or hydrochloride of amino acid ester (0.15 mmol, 1.5 equiv.), Et$_3$N (0.2 mmol, 2.0 equiv.), and DCM (2.0 mL) were added sequentially. The vial was capped, and stirring for 48 h in an oil bath at 60 °C. After completion of the reaction (monitored by TLC), the solvent was evaporated under reduced pressure, and the residue was purified by flash chromatography on silica gel (Hexanes: EtOAc = 10:1 to 1:1) to afford a ring opening amide products.

### General procedure for the synthesis of ester products 50−52

To a 10-mL oven-dried vial containing a magnetic stirring bar, and sodium hydride (60% dispersion in mineral oil, 0.2 mmol, 2.0 equiv.) in dry THF (2.0 mL), was added alcohol (0.15 mmol, 1.5 equiv.) dropwise under stirring at 0 °C under a nitrogen atmosphere. After the reaction mixture turned to clear, a solution of cyclobutanone **2** (0.1 mmol) in THF (1.0 mL) was added dropwise at room temperature, then the reaction was stirred at 60 °C for 0.5 h. Saturated aqueous NH$_4$Cl (10 mL) was added to quench the reaction, the organic phase was separated, and the aqueous layer was extracted with EtOAc (3 × 5.0 mL). The combined organic layer was washed with brine, dried over anhydrous MgSO$_4$, and concentrated in vacuo after filtration. The residue was purified by chromatography on silica gel (Hexanes: EtOAc = 30:1) to afford the ester products.

**Fig. 7 | Control experiments. a** Control reaction of **1af** in the presence of styrene under standard conditions. **b** Control reaction of **1af** in the presence of BnOH under standard conditions. **c** Ring-opening reaction of **70** with methyl L-tryptophanate.

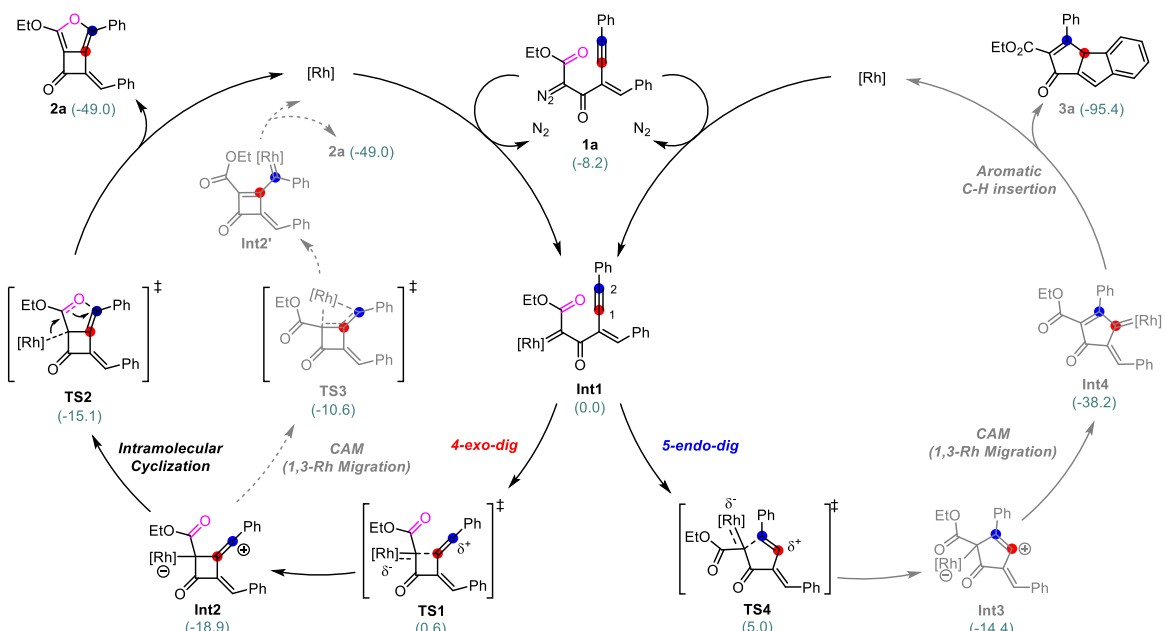

**Fig. 8 | The reaction mechanism and DFT calculation for the 4-exo-dig and 5-endo-dig carbocyclization using Rh$_2$(HCOO)$_4$ as catalyst.** The numbers in the parentheses are the free energies in kcal/mol, computed at SMD(EtOAc)/BMK/def2-TZVPP//B3LYP/def2-SVP, all the structures are obtained in the gas phase.

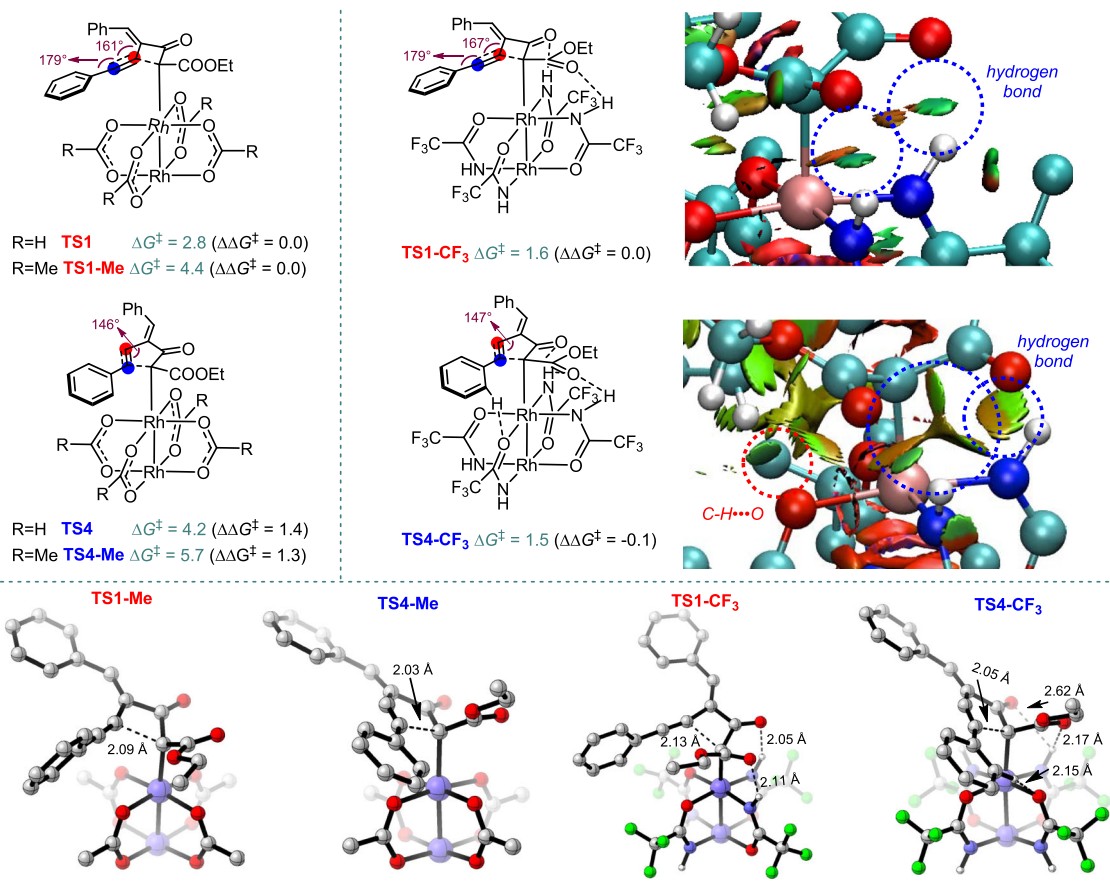

**Fig. 9 | The key transition states in the regioselectivity of the real system and the NCI plots.** Free energies are in kcal/mol, computed at SMD(EtOAc)/BMK/def2-TZVPP//SMD(EtOAc)/B3LYP/def2-SVP, all the structures are obtained in solution.

## General procedure for the synthesis of benzocyclobutenones 59–62

To a 10-mL oven-dried vial containing a magnetic stirring bar, **2** (0.2 mmol), maleimide (0.3 mmol, 1.5 equiv.), and mixed solvent toluene/DCE (1.0/0.1 mL) were added in sequence, and the reaction mixture was stirred for 12 h at 90 °C. Upon completion, the mixture was cooled to room temperature, and the solvent was removed under reduced pressure. The residue was purified by silica gel chromatography (Hexanes: EtOAc = 10:1 to 5:1) to afford pure benzocyclobutenone products.

## General procedure for the synthesis of poly-substituted benzene derivatives 63–69

To a 10-mL oven-dried vial containing a magnetic stirring bar, the above obtained benzene fused cyclobutanone derivative (0.1 mmol), piperidine (0.15 mmol, 1.5 equiv.), and DCM (2.0 mL) were added sequentially. The vial was capped, and stirring for 24 h in an oil bath at 60 °C. After completion of the reaction (monitored by TLC), the solvent was evaporated under reduced pressure, and the residue was purified by flash chromatography on silica gel (Hexanes: EtOAc = 10:1 to 1:1) to afford the ring opening products.

## Data availability

The data supporting the findings of this study are available within the paper and its Supplementary Information. Crystallographic data for the structures reported in this Article have been deposited at the Cambridge Crystallographic Data Center, under deposition numbers CCDC 2249373 (2g), 2246883 (2af), 2110463 (38), and 2083438 (59),

and 2265899 (71). Copies of the data can be obtained free of charge via https://www.ccdc.cam.ac.uk/structures/.

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

## Acknowledgements

Support for this research from the National Natural Science Foundation of China (21971262), Guangdong Provincial Key R&D Programme (21202107201900002) and Guangdong Basic and Applied Basic Research (2021A1515010384) is greatly acknowledged.

## Author contributions

Z.-X.Y. and X.-F.X. conceived and supervised the project, and wrote the manuscript. K.H. and Y.Z. designed the experiments. K.H., H.Y. and S.D. performed the experiments and analyzed the data. Z.-X.Y. and Y.Z. conducted the DFT calculation. Z.Z. and J.H. carried out the anticancer activity evaluation. W.H. supervised the project and revised the manuscript. All authors discussed the results and commented on the article.

## Competing interests

The authors declare no competing interests.
