## [Peer Review File · Nature Communications]

REVIEWER COMMENTS

Reviewer #1 (Remarks to the Author):

The manuscript disclosed an efficient and direct assembly of aromatic ring-fused cyclobutanones beyond benzocyclobutenone (BCB) skeletons through an unprecedented Rh-catalyzed 4-exo-dig carbocyclization process, which featured high to excellent yields and good substrate tolerance. Further transformation of synthesized furan-fused cyclobutanones into valuable furan-embedded amide and ester derivatives highlighted their potential application in synthetic and medicinal chemistry, resulting in several compounds showing significant and broad anticancer potency against different human cancer cell lines at micromolar concentrations. Significantly, the author proved that 4-exo-dig carbocyclization is mainly cyclization step by lowering angle strain of the key sp²-hybridized vinyl cationic transition state, instead of the competitive 5-endo-dig process. In summary, this research involves a large amount of work and novelty in the structure of the target product. However, there several issues need to be addressed before the publication as follows:

- 1) Compared with benzocyclobutenone, the authors have emphasized the methodological usefulness of this study in preparing aromatic ring-fused cyclobutanones, such as furan-fused cyclobutanone. This naturally leads to an important topic of whether this strategy is also suitable for constructing thiophene-fused or pyrrole-fused cyclobutanone. If so, these results will significantly break the limitations in this field.
- 2) The author also demonstrated the application of furan-embedded amide and ester derivatives in late-stage functionalization with the addition of different nucleophilic substrates, such as primary/secondary amines and bioactive products containing a free hydroxyl group. Considering that the authors have screened out several molecules with good anticancer activity, the reviewer was curious if this method works well by using drug-like substrates with reactive carbon atom. If so, this will significantly increase the diversity of target products and further facilitate the discovery of new lead compounds.
- 3) Meanwhile, dose the author explore chemical reactions at electron-deficient olefin reaction sites of cyclobutanone moiety, although it is not favorable? Most likely, the utilization of substrates with reactive carbon sites may bring a surprise. Of course, sufficient data are needed to support the possible reaction pathway.
- 4) Moreover, the reviewer found that products 9-58 had the characteristics of axial chiral compounds. As a hot topic in synthetic chemistry, the author is suggested to explore the asymmetric transformation of these compounds for constructing axially chiral styrenes because these novel structures may facilitate the discovery of new catalysts and ligands.
- 5) Booking a professional agency's language polishing service will improve the quality of the article even if the existing language is ok.

In summary, this is a well-designed and organized research work. However, considering the high standard and novelty required for research content in Nature Communications, the reviewer recommends the publication for other journals after minor revisions, such as JACS or ACIE.

Reviewer #2 (Remarks to the Author):

The manuscript submitted is of high quality, and the results presented are remarkable. It describes a cyclization in which a Rh (Fischer-type) carbene adds electrophilically to a proximate C-C triple bond in a 4-exo (rather than 5-endo) manner. The cyclic 4-membered intermediates formed undergo further [3+2] cyclization to produce furan fused 2-alkylidene cyclobutanones **2** in high yields, with a broad scope. The products **2** were then converted in a variety of interesting derivatives (via known reactions), some of them showing significant anticancer activity.

An interesting part of the manuscript is the reaction mechanism. In particular, using Rh₂(tfacam)₄ (bearing an amide N-H group) as catalyst, expected 5-endo-dig cyclization takes place through a CAM type mechanism followed by aromatic C-H insertion, with formation of five membered lactones **3** together with a minor amount of **2**. Using Rh₂(OAc)₄ or also Rh₂(OPiv)₄, unusual 4-exo-dig cyclization occurs with high selectivity, and subsequent cyclization affords final products **2**. An accurate DFT calculation supports the experimental data, and the prevalence of the 4-exo-dig process was attributed to the angle strain in vinyl carbocation intermediate formed in the competitive 5-endo-dig process. On the other hand, with Rh₂(tfacam)₄ catalyst an intermolecular hydrogen bond between the N-H of tfacam and a C=O in transition states lowers the activation energies for both 4-exo and 5-endo paths, making these energies nearly similar, with loss of selectivity.

To decide if this contribution deserves publication in Nat. Comm. (i.e. a top level interdisciplinary scientific journal) the key point are the importance and general significance of 4-exo-dig cyclization of alkynes. It should be noted that this process, although rare, was observed in various cases, namely radical or anionic cyclizations, as reported in Ref. 60. Also, the following ref. (Alcaide & coll, Chem Commun, 2011, 47, 9054) that describes a gold-catalyzed 4-exo-dig allene cyclization, should be considered.

Ref 60 at pag 6531 states:

"Electrophilic cyclization (of alkynes): As discussed above, the

4-exo/5-endo competition is much closer than it is for the 3-exo/

4-endo and 5-exo/6-endo pairs of cyclizations, owing largely to

the strain destabilization of only one (the four-membered) of the two products. However, the situation changes in cationic closures where the 5-endo product would be highly strained as well because of the inclusion of an sp-hybridized cationic center in the ring. The instability of vinyl cations is the likely reason why electrophilic 4-exo/5-endo closures are not described in the literature..."

Therefore, one valuable result in this manuscript is that such cyclization takes place; this is probably due to the presence of Rh and also to the conjugation of vinyl cation (Int. 1) with Ar. However can this process be considered general? What would happen if in substrates 1 there was an alkyl group instead of Ar?

To sum up, the prevalence of 4-exo-dig over 5-endo-dig is not enough for considering the publication in Nat. Commun. However I evaluate that this contribution is of good level, and suitable for other important chemical journals.

Minor changes:

- Reaction times should be indicated in Tables
- Yields of substrates 1 should be indicated in Supplementary Information
- In the description of NMR spectra please use "broad" or "bs" when appropriate (OH, NH)

Reviewer #4 (Remarks to the Author):

Manuscript NCOMMS-22-50609-T by Zhi-Xiang Yu, Xinfang Xu, and co-workers describes the cyclization of alkyne tethered diazocompounds under rhodium catalysis for the synthesis of furan-fused cyclobutanones. This new method relies on a known reactivity, namely the carbocyclization of alkynes, but by judiciously designing a tether that conformationally constrains the substrate, the authors can trigger a reaction that allows for the unprecedented formation of a cyclobutanone, a small and strained ring. The significance of the work stems from the excellent control of the selectivity of this powerful reaction towards the most strained of the possible regioisomers, which is also an extremely interesting scaffold.

The study is complemented by an analysis of the synthetic applications of the furan-fused cyclobutanone products, studying the nucleophilic opening of the cyclobutanone with an excessively

high number of examples on the opening with alcohols and amines, and very interestingly showing the potential to forge benzocyclobutenones by a Diels-Alder/aromatization sequence. Then follows an evaluation of the anticancer activity of some selected compounds, a part that is outside the scope of my expertise. And finally, the mechanism of the transformation is fully elucidated by DFT calculations giving insight into the factors that define the regioselectivity of the carbocyclization. Understanding the factors that lie behind the selectivity observed can have a great significance in the field.

The presentation of results, the methodology, the claims made, and the conclusions reached are generally sound however some points require revision. Below is a summary of the identified weaknesses:

- The presentation of the previous results and the context of the project is comprehensive, although Figure 1b is misleading. It should be redesigned to make it clear that the size of the ring that can be obtained depends on the tether length. On redrafting the Scheme please also include silver as one of the metals that can catalyze this type of reactivity.

- The methodology used in the optimization of the reaction is sound although the authors do not run a blank experiment for the reaction in Table 1, that is key to know if the transition metal catalyst is necessary for the transformation.

- On studying the scope of the 4-exo-dig carbocyclization reaction (table 2), the authors fail to consider key parameters such as:

- o Is it possible to react a terminal alkyne? And a TMS substituted one?

- o Is it possible to change the geometry of the styryl moiety?

- o Is it possible to change the ester group to a ketone, aldehyde, or amide?

- Do the authors have an explanation as to why 44 is a cyclic valerolactam (not a caprolactam as stated in the text, and not product 45 as also incorrectly stated in the text), whereas in the nucleophilic opening leading to 43 cyclization to the caprolactam does not occur?

- In the formation of benzocyclobutenones, a better description of the process, namely a Diels-Alder reaction followed by aromatization, should be included and this new methodology toward benzocyclobutenones should be compared to the ones already described in the literature.

- The authors should justify their chosen level of theory. Do they have benchmark data to support their choice?

- The formation of the rhodium carbene is not computed. This is likely to be the rate determining step of the overall transformation (if not the reaction should occur at lower temperature).

- In the 4-exo-dig transformations, the authors labelled the transformation from Int2 to 2a as a "Formal [3+2] cyclization". In fact, it is the transformation from Int1 to 2a that would be a formal stepwise [3+2] cyclization. The step from Int2 to 2a is better described as an intramolecular cyclization. In addition, the authors should consider if the transformation from Int1 to Int2 can be labelled as a CAM since the carbenic character is not transferred to an alkyne carbon.

- The authors should justify why in Figure S4 the energy for 3TS5 is lower than the energy for 3Int3.
- Rh₂(OAc)₄ and Rh₂(tfacam)₄ are referred to as the two real catalysts. None of them is used in the optimized reaction. A better explanation on the choice of these catalytic systems for the study of the catalyst effects on the regiochemistry should be provided.
- Whereas the calculated energies for the 4-exo-dig and 5-endo-dig cyclizations using Rh₂(OAc)₄ as a catalyst provide a fairly good explanation for the results observed, this is not the case for the Rh₂(tfacam)₄. Have the authors considered the effect that the axial coordination of a solvent on the dirhodium core might have (see for instance *Organometallics* 2021, 40, 4120–4132)? Ethyl acetate is a quite exotic solvent for metal carbene transformations that might have a non-negligible effect.
- The spectra reported in the supporting information file have very low intensity for the peaks and this does not allow determining the purity of the synthesized products.

The manuscript by Hong and coworkers focuses on synthesizing a variety of achiral furan-fused cyclobutanones using carbene-alkyne metathesis (CAM), which the authors called a 4-exo-dig-cyclization approach. The furan-fused cyclobutanones are then converted to various analogs using post-modification. Some analogs are also evaluated for their anticancer activities against several human cancer cell lines. The authors also performed computational studies to support their proposed mechanistic hypothesis. Overall, this work is an unwarranted combination of unrelated bits and pieces combined in a single manuscript to increase the content for a prestigious journal such as Nature Communication. I have reservations about the suitability of this work as a communication in any scientific journal in its current form. To sustain my arguments, I will point out the following weaknesses of this work in the present form.

- 1) The substrate scope of the carbene-alkyne metathesis (CAM) to synthesize furan-fused cyclobutanones is very limited. The authors utilize a specific-substrate design for the presented CAM, which only leads to a limited number of achiral cyclobutanones. To increase the content of the work, the authors then performed trivial post-modifications to access more than 50 analogs. Overall, there is a lack of innovation in this manuscript, and there was no need for too many similar transformations to increase the number of examples. I also have reservations about invoking the nomenclature of Baldwin's ring closure cyclization (4-exo-dig or 5-endo-dig) in this carbene-alkyne metathesis.
- 2) In the manuscript, there is a section called mechanistic investigations. Unfortunately, the authors didn't perform any experiments to probe the mechanism but came up with a proposed mechanism and performed DFT calculations to justify the product 2a vs 3a, which they called 4-exo-dig vs. 5-endo-dig cyclization. In the literature, there is no example of invoking Baldwin's rules in carbene-alkyne metathesis reactions. Furthermore, 5-endo-dig cyclizations using carbene-alkyne metathesis are not possible due to unfavorable trajectories. Still, the authors invoke a 5-endo-dig cyclization to justify the formation of product 3a and performed DFT calculations. I suggest a different mechanistic pathway for the formation of product 3a from the same intermediate used to form product 2a.

Figure 1: Plausible Mechanism based on the bond angles and trajectories

This is just a suggestion for consideration.

- 3) The authors evaluated several analogs for anticancer activities against human cancer cell lines. I fail to understand the logic behind these biological studies and what the authors want to communicate with the weak μM IC_{50} values.
- 4) Minor Point: There are satellite peaks in multiple ^1H spectra. Also, the peak height for several ^1H and ^{13}C spectra is too low in many compounds in the supporting information. I suggest the authors reprocess the NMR spectra to make peaks visible for each functionality.

With all these considerations above and the lack of novelty in this work, I reject this manuscript in its present form for publication in Nature Communications keeping in mind the readership of this prestigious journal.

For Reviewer #1 (Remarks to the Author):

The manuscript disclosed an efficient and direct assembly of aromatic ring-fused cyclobutanones beyond benzocyclobutenone (BCB) skeletons through an unprecedented Rh-catalyzed 4-exo-dig carbocyclization process, which featured high to excellent yields and good substrate tolerance. Further transformation of synthesized furan-fused cyclobutanones into valuable furan-embedded amide and ester derivatives highlighted their potential application in synthetic and medicinal chemistry, resulting in several compounds showing significant and broad anticancer potency against different human cancer cell lines at micromolar concentrations. Significantly, the author proved that 4-exo-dig carbocyclization is mainly cyclization step by lowering angle strain of the key sp-hybridized vinyl cationic transition state, instead of the competitive 5-endo-dig process. In summary, this research involves a large amount of work and novelty in the structure of the target product. However, there several issues need to be addressed before the publication as follows:

We appreciate the reviewer for the suggestions and comments.

1) Compared with benzocyclobutenone, the authors have emphasized the methodological usefulness of this study in preparing aromatic ring-fused cyclobutanones, such as furan-fused cyclobutanone. This naturally leads to an important topic of whether this strategy is also suitable for constructing thiophene-fused or pyrrole-fused cyclobutanone. If so, these results will significantly break the limitations in this field.

Thank you for the suggestion, we have tried to prepare the materials for the synthesis of corresponding thiophene-fused or pyrrole-fused cyclobutanones. To our delight, pyrrole-fused cyclobutanone 2ae was formed in 42% yield when triazole 1ae was used as the carbene precursor in the presence of Rh₂(esp)₂ in DCE at 80 °C (Fig. S1a); whereas, the preparation of the material for the synthesis of thiophene-fused cyclobutanone was failed with reported methods (Fig. S1b). However, it's predictable that the thiophene-fused cyclobutanone should be formed when the corresponding diazo compound is accessible.

Fig. S1 | Synthesis of thiophene-fused and pyrrole-fused cyclobutanones.

In addition, we also further expanded the reaction scope with additional 13 examples of compounds with different functionalities, which are listed below (Fig. S2). All these results have been added to Table 2 in the revised manuscript with discussion, and corresponding data have been added to the updated SI.

Fig. S2 | Newly added examples for the synthesis of furan-fused cyclobutanones.

2) The author also demonstrated the application of furan-embedded amide and ester derivatives in late-stage functionalization with the addition of different nucleophilic substrates, such as primary/secondary amines and bioactive products containing a free

hydroxyl group. Considering that the authors have screened out several molecules with good anticancer activity, the reviewer was curious if this method works well by using drug-like substrates with reactive carbon atom. If so, this will significantly increase the diversity of target products and further facilitate the discovery of new lead compounds.

Thank you for the suggestion, we have tried using indole and indolone as the nucleophiles, which are privileged motifs in drug molecules, for this ring-opening transformation, however, no reaction occurred (Fig. S3a and S3b).

However, when the reaction using trimethylsulfoxonium iodide as a reactive carbonic nucleophile precursor, the ring-opening product 7 was generated in 50% yield in the presence of NaH (Fig. S3c). Moreover, when diazoamide was used as the reagent, which is supposed to form the zwitterionic intermediate in situ in the presence of gold catalyst, the ring-opening adduct 8 was formed in 56% yield (Fig. S3d).

These results have been added to Fig 2 in the revised manuscript with discussion, and the corresponding data have been added to the SI.

Fig. S3 | Ring-opening reaction using carbonic nucleophiles.

3) Meanwhile, does the author explore chemical reactions at electron-deficient olefin reaction sites of cyclobutanone moiety, although it is not favorable? Most likely, the utilization of substrates with reactive carbon sites may bring a surprise. Of course, sufficient data are needed to support the possible reaction pathway.

Thank you for the suggestion. We have tried a few of other different nucleophiles, including Grignard reagent and few other reactive carbonic nucleophiles, and the corresponding reactions all occurred on the carbonyl group, which might be due to the increased reactivity aroused by the strained four-membered ring structure, and

this is also consistent with previously reported reactivity of benzocyclobutenone. Corresponding comment has been added to the revised manuscript, and the new reference has been cited as ref 60: Bradley, J., Durst, T. & Williams A. J. Thermolysis of 2-benzylidenebenzocyclobutenols. J. Org. Chem. 57, 6575-6579 (1992)

For the possible reaction pathway, we have provided experimental proof with additional density functional theory (DFT) calculations to sustain the proposed reaction mechanisms involving cationic intermediate (see Fig. S4 below for details).

Fig. S4 | Control experiments.

Initially, the reaction of **1af** in the presence of excess amount of styrene under standard conditions was carried out, which intended to obtain the cyclopropanation product with possible carbene intermediate(s). However, only furan-fused cyclobutanone **2af** was formed in 71% yield, and no cyclopropane product was observed by mass spectrometry (MS) analysis of the crude reaction mixture (**Fig. S4a**), which implies that the intramolecular cascade reaction might be more favorable even if the second carbene species was formed *via* the CAM process. On the contrary, the corresponding addition product with BnOH was isolated in 31% yield combined with **2af** in 40% yield (**Fig. S4b**), and this cyclobutanone product **70** was confirmed by X-ray crystallography after converting to the ring-opening product **71** (**Fig. S4c**). These results suggested potential carbene or cationic intermediates in this cascade

reaction, and the density functional theory (DFT) calculations support the cationic intermediate (see **Fig. S5** below details).

Fig. S5 | Competition between intramolecular cyclization and intermolecular O-H insertion of substrate **1af in EtOAc.** Computed at SMD(EtOAc)/ ω B97X-D/def2-TZVPP//B3LYP/def2-SVP.

According to the DFT studies, **Int2-CP** is obtained through 4-*exo-dig* cyclization from the substrate **1af** and $\text{Rh}_2(\text{OAc})_4$. Then **Int2-CP** can undergo intramolecular cyclization *via* **TS2-CP**, which will deliver product **2af**, requiring activation free energy of 3.5 kcal/mol (path a). The 1,3 Rh-migration, which forms Rh-carbene *via* **TS3-CP**, is disfavored over **TS2-CP** by 6.7 kcal/mol (path b). When BnOH is added, the vinyl cation **Int2-CP** can undergo O-H insertion to afford **Int3-CP** *via* **TS4-CP**. This process is highly exothermic (-30.6 kcal/mol) and has comparable activation free energies with **TS2** (3.5 kcal/mol vs. 3.5 kcal/mol), which is consistent with our experiments. Then **Int3-CP** can afford the O-H insertion product through sequential proton transfer process. DFT calculations suggested that direct 1,5-proton transfer *via* **TS5-CP** is difficult, requiring activation free energy of 28.2 kcal/mol. However, with the assistant of an additional molecule of BnOH as a proton transporter, the proton transfer is facilitated. This process requires activation free energy of 22.4 kcal/mol *via* **TS6-CP**, affording the O-H insertion product **70**.

All these results have been added to Fig. 6 in the revised manuscript with discussions, and the DFT study of these reaction outcomes has been add to Fig. S8 in SI with discussions. All these results are in consistent with proposed reaction pathway involving cationic intermediate.

4) Moreover, the reviewer found that products 9-58 had the characteristics of axial chiral compounds. As a hot topic in synthetic chemistry, the author is suggested to explore the asymmetric transformation of these compounds for constructing axially chiral styrenes because these novel structures may facilitate the discovery of new catalysts and ligands.

Thank you for the suggestion, we have conducted the ring-opening reaction with L-proline and D-proline, and only one stereoisomer was isolated for each case and with the same proton NMR spectra (see Fig. S6 below for detail), which indicate that this reaction either has very high diastereoselectivity (if there is a newly formed axial chirality) or does not have a newly formed axial chirality.

Furthermore, we also did the DFT calculation at B3LYP/def2-SVP level, and the calculated activation free energy of the single bond rotation is 8.5 kcal/mol (Fig. S7), indicating that the racemization of this compound is facile. Thus, the axial chirality in the present trisubstituted substrate might not exist.

Fig. S6 | Proton NMR spectra of the ring-opening products derived from L-/D-proline.

Fig. S7 | DFT calculation study of the activation free energy of the single bond rotation of the ring-opening product.

5) Booking a professional agency's language polishing service will improve the quality of the article even if the existing language is ok.

Thank you for the suggestion, we have sent this manuscript to the experts who are native English speaker for the language revision, including Prof. Michael P. Doyle from UTSA who is an expert in carbene chemistry.

For Reviewer #2 (Remarks to the Author):

The manuscript submitted is of high quality, and the results presented are remarkable. It describes a cyclization in which a Rh (Fischer-type) carbene adds electrophilically to a proximate C-C triple bond in a 4-exo (rather than 5-endo) manner. The cyclic 4-membered intermediates formed undergo further [3+2] cyclization to produce furan fused 2-alkylidene cyclobutanones **2** in high yields, with a broad scope. The products **2** were then converted in a variety of interesting derivatives (via known reactions), some of them showing significant anticancer activity.

An interesting part of the manuscript is the reaction mechanism. In particular, using Rh₂(tfacam)₄ (bearing an amide N-H group) as catalyst, expected 5-endo-dig cyclization takes place through a CAM type mechanism followed by aromatic C-H insertion, with formation of five membered lactones **3** together with a minor amount of **2**. Using Rh₂(OAc)₄ or also Rh₂(OPiv)₄, unusual 4-exo-dig cyclization occurs with high selectivity, and subsequent cyclization affords final products **2**. An accurate DFT calculation supports the experimental data, and the prevalence of the 4-exo-dig process was attributed to the angle strain in vinyl carbocation intermediate formed in the competitive 5-endo-dig process. On the other hand, with Rh₂(tfacam)₄ catalyst an intermolecular hydrogen bond between the N-H of tfacam and a C=O in transition

states lowers the activation energies for both 4-exo and 5-endo paths, making these energies nearly similar, with loss of selectivity.

To decide if this contribution deserves publication in Nat. Comm. (i.e. a top level interdisciplinary scientific journal) the key point are the importance and general significance of 4-exo-dig cyclization of alkynes. It should be noted that this process, although rare, was observed in various cases, namely radical or anionic cyclizations, as reported in Ref. 60. Also, the following ref. (Alcaide & coll, Chem Commun, 2011, 47, 9054) that describes a gold-catalyzed 4-exo-dig allene cyclization, should be considered.

We appreciate the reviewer for the suggestions and comments. And corresponding reference has been cited in the revised manuscript as ref 64:

Alcaide, B., Almendros, P., Campo, T. M. & Fernández, I. Fascinating reactivity in gold catalysis: synthesis of oxetenes through rare 4-exo-dig allene cyclization and infrequent β -hydride elimination. *Chem. Commun.* **47**, 9054-9056 (2011).

1) Ref 60 at pag 6531 states: "Electrophilic cyclization (of alkynes): As discussed above, the 4-exo/5-endo competition is much closer than it is for the 3-exo/4-endo and 5-exo/6-endo pairs of cyclizations, owing largely to the strain destabilization of only one (the four-membered) of the two products. However, the situation changes in cationic closures where the 5-endo product would be highly strained as well because of the inclusion of an sp-hybridized cationic center in the ring. The instability of vinyl cations is the likely reason why electrophilic 4-exo/5-endo closures are not described in the literature" Therefore, one valuable result in this manuscript is that such cyclization takes place; this is probably due to the presence of Rh and also to the conjugation of vinyl cation (Int. 1) with Ar. However, can this process be considered general? What would happen if in substrates 1 there was an alkyl group instead of Ar?

Thank you for the comments and suggestion, the substrate 1 with an alkyl group instead of Ar has been subjected to the current conditions, and gave the product 2u in 64% yield.

In addition, we also further expanded the reaction scope with additional 13 examples of compounds with different functionalities (see Fig. S2 above for the response to the first reviewer's first question for details), especially for the synthesis of pyrrole-fused one 2ae, spiro[2.3]hexan-4-one 2af, and ketone derived product

2ag. All these results are added to Table 2 with discussion, and the corresponding data are included in SI.

2) Reaction times should be indicated in Tables

Thank you for the suggestion, the corresponding revision has been made in the revised manuscript.

3) Yields of substrates 1 should be indicated in Supplementary Information

Thank you for the suggestion, the corresponding yields for the synthesis of 1 have been added in the SI.

4) In the description of NMR spectra please use “broad” or “bs” when appropriate (OH, NH)

Thank you for the suggestion, corresponding revision has been made in the SI.

For Reviewer #3 (Remarks to the Author):

The manuscript by Hong and coworkers focuses on synthesizing a variety of achiral furan-fused cyclobutanones using carbene-alkyne metathesis (CAM), which the authors called a 4-exo-dig-cyclization approach. The furan-fused cyclobutanones are then converted to various analogs using post-modification. Some analogs are also evaluated for their anticancer activities against several human cancer cell lines. The authors also performed computational studies to support their proposed mechanistic hypothesis. Overall, this work is an unwarranted combination of unrelated bits and pieces combined in a single manuscript to increase the content for a prestigious journal such as Nature Communication. I have reservations about the suitability of this work as a communication in any scientific journal in its current form. To sustain my arguments, I will point out the following weaknesses of this work in the present form.

We appreciate the reviewer for the comments.

Cyclobutanone is a strained all-carbon motif that is present in many natural products, bioactive molecules, and materials. Although a variety of catalytic

strategies have been documented for the construction of this four-membered cyclic ketone, direct assembly of the aromatic ring fused cyclobutanones beyond benzocyclobutenone (BCB) skeletons remains an uncharted area in synthetic chemistry. In this work, we report a practical and convenient method for the direct construction of furan-fused cyclobutanone, which is an unknown molecule and has not yet been prepared, providing an intriguing opportunity for the discovery of untapped properties and reactivities.

1) The substrate scope of the carbene-alkyne metathesis (CAM) to synthesize furan-fused cyclobutanones is very limited. The authors utilize a specific-substrate design for the presented CAM, which only leads to a limited number of achiral cyclobutanones. To increase the content of the work, the authors then performed trivial post-modifications to access more than 50 analogs. Overall, there is a lack of innovation in this manuscript, and there was no need for too many similar transformations to increase the number of examples. I also have reservations about invoking the nomenclature of Baldwin's ring closure cyclization (4-exo-dig or 5-endo-dig) in this carbene-alkyne metathesis.

We appreciate the reviewer for the suggestions. We further expanded the reaction scope with additional 13 examples of compounds with different functionalities (see Fig. S2 above for the response to the first reviewer's first question for details), especially for the synthesis of pyrrole-fused one 2ae, spiro[2.3]hexan-4-one 2af, and ketone derived product 2ag. All these results are added to Table 2 with discussion, and the corresponding data are included in SI.

These previously inaccessible and unknown furan-fused cyclobutanone scaffolds, which could be served as a platform molecule, participating in different types of ring-opening and cycloaddition reactions as versatile synthetic building blocks for diversity-oriented-synthesis. A variety of drugs and sterols are tolerated in the ring-opening transformation of these synthesized furan-fused cyclobutanones under mild conditions, demonstrating the potent synthetic value of this method. Thus, this method might be used as a potential click reaction for the late-stage modification of complex molecules and in bioorthogonal chemistry. Further synthetic applications are under exploration and will be reported in due course.

2) In the manuscript, there is a section called mechanistic investigations. Unfortunately, the authors didn't perform any experiments to probe the mechanism but came up with a proposed mechanism and performed DFT calculations to justify the product 2a vs 3a, which they called 4-exo-dig vs. 5-endo-dig cyclization. In the literature, there is no example of invoking Baldwin's rules in carbene-alkyne metathesis reactions. Furthermore, 5-endo-dig cyclizations using carbene-alkyne metathesis are not possible due to unfavorable trajectories. Still, the authors invoke a 5-endo-dig cyclization to justify the formation of product 3a and performed DFT calculations. I suggest a different mechanistic pathway for the formation of product 3a from the same intermediate used to form product 2a.

Figure 1: Plausible Mechanism based on the bond angles and trajectories

We appreciate the reviewer for the comments and suggestions.

According to the Baldwin's rule and its revised version on the alkyne cyclization (see Chem. Rev. 2011, 111, 6513–6556), both the electrophilic 5-endo-dig and 4-exo-dig cyclizations are at borderlines rather than “unfavorable”, but these two cyclizations would have comparable activation free energies due to the ring strains in the four-membered ring and 5-endo-dig product with an internal vinyl cation (see part 3.2 in Chem. Rev. 2011, 111, 6513–6556). These predictions are consistent with our computations and both cyclizations are observed in our experiments using an appropriate substrate/condition.

For the alternative pathway for the formation of 3a, we have considered about the plausible mechanism ([1+2] cyclization) as suggested. Unfortunately, the corresponding transition state and 3,4-fused bicyclic intermediate cannot be located. This is reasonable because this intermediate has large ring strains with a fused cyclopropene and cyclobutanone motif. Thus, the red and blue C-C bonds in the fused structure are easily broken, and we can only locate Int2/Int3 and cyclization transition states TS1/TS4 (see Fig. S8: e.g., the following optimization job gave Int3

directly). We have mentioned this result in the revised manuscript and the corresponding data have been updated in SI.

Fig. S8 | The DFT calculation study of other possible reaction pathway.

In addition, we have provided experimental proof with additional density functional theory (DFT) calculations to sustain the proposed reaction mechanisms involving cationic intermediate (see Fig. S4 and S5 above for the response to the third question of the first reviewer for details). These results have been added to Fig. 6 in the revised manuscript with discussions, and the DFT study of these reaction outcomes has been add to Fig. S8 in SI with discussions. All these results are in consistent with proposed reaction pathway involving cationic intermediate.

3) The authors evaluated several analogs for anticancer activities against human cancer cell lines. I fail to understand the logic behind these biological studies and what the authors want to communicate with the weak μM IC_{50} values.

We appreciate the reviewer for the comments. The preliminary antitumor activity study of these synthesized products indicated that some of these unique molecules exhibited significant and broad anticancer potency against different human cancer cell lines (29: HCT116 cells, $\text{IC}_{50} = 1.19 \mu\text{M}$; A549 cells, $\text{IC}_{50} = 2.74 \mu\text{M}$; SJSA-1 cells, $\text{IC}_{50} = 2.08 \mu\text{M}$; 50: MCF-7 cells, $\text{IC}_{50} = 4.23 \mu\text{M}$; MDA-MB-231 cells, $\text{IC}_{50} = 6.88 \mu\text{M}$). Further applications, including but not limited to novel transformations

development and bioactive molecule discovery, could be expected in due course.

4) Minor Point: There are satellite peaks in multiple ¹H spectra. Also, the peak height for several ¹H and ¹³C spectra is too low in many compounds in the supporting information. I suggest the authors reprocess the NMR spectra to make peaks visible for each functionality.

We appreciate the reviewer for the suggestion, and corresponding revision has been made in SI.

Reviewer #4 (Remarks to the Author):

Manuscript NCOMMS-22-50609-T by Zhi-Xiang Yu, Xinfang Xu, and co-workers describes the cyclization of alkyne tethered diazocompounds under rhodium catalysis for the synthesis of furan-fused cyclobutanones. This new method relies on a known reactivity, namely the carbocyclization of alkynes, but by judiciously designing a tether that conformationally constrains the substrate, the authors can trigger a reaction that allows for the unprecedented formation of a cyclobutanone, a small and strained ring. The significance of the work stems from the excellent control of the selectivity of this powerful reaction towards the most strained of the possible regioisomers, which is also an extremely interesting scaffold.

The study is complemented by an analysis of the synthetic applications of the furan-fused cyclobutanone products, studying the nucleophilic opening of the cyclobutanone with an excessively high number of examples on the opening with alcohols and amines, and very interestingly showing the potential to forge benzocyclobutenones by a Diels-Alder/aromatization sequence. Then follows an evaluation of the anticancer activity of some selected compounds, a part that is outside the scope of my expertise. And finally, the mechanism of the transformation is fully elucidated by DFT calculations giving insight into the factors that define the regioselectivity of the carbocyclization. Understanding the factors that lie behind the selectivity observed can have a great significance in the field.

The presentation of results, the methodology, the claims made, and the conclusions reached are generally sound however some points require revision. Below is a summary of the identified weaknesses:

We appreciate the reviewer for the suggestions and comments.

1) The presentation of the previous results and the context of the project is comprehensive, although Figure 1b is misleading. It should be redesigned to make it clear that the size of the ring that can be obtained depends on the tether length. On redrafting the Scheme please also include silver as one of the metals that can catalyze this type of reactivity.

Thank you for the suggestion. Corresponding revision has been made in Fig 1 as suggested. Moreover, silver is added as one of the metals that can catalyze this type of reactivity, and corresponding recent reference has been cited as ref 51:

Díaz-Jiménez, À., Monreal-Corona, R., Poater, A., Álvarez, M., Borrego, E., Pérez, P. J., Caballero, A., Roglans, A. & Pla-Quintana, A. Intramolecular interception of the remote position of vinylcarbene silver complex intermediates by C(sp³)-H bond insertion. *Angew. Chem. Int. Ed.* **62**, e202215163 (2023).

2) The methodology used in the optimization of the reaction is sound although the authors do not run a blank experiment for the reaction in Table 1, that is key to know if the transition metal catalyst is necessary for the transformation.

Thank you for the suggestion. Corresponding control reaction has been conducted, and no reaction occurred in the absence of metal catalyst under otherwise identical conditions. These results have been mentioned in the revised manuscript with discussion in the optimization part.

3) On studying the scope of the 4-exo-dig carbocyclization reaction (table 2), the authors fail to consider key parameters such as:

Is it possible to react a terminal alkyne? And a TMS substituted one?

Thank you for the suggestion. We have prepared the suggested materials 1ai and 1aj, and tested under the current conditions. However, only the eneyne product 2ai was isolated in 34% yield when TMS protected material 1ai was used, which formed via a Wolff rearrangement, nucleophilic addition with H₂O, and decarbonation sequence. Whereas, only decomposition of the terminal alkyne 1aj was observed under current conditions (Fig. S9). These results have been mentioned in the revised manuscript with discussion.

In addition, we further expanded the reaction scope with additional 13 examples of compounds with different functionalities (see Fig. S2 above for the response to the

first reviewer's first question for details). All these results are added to Table 2 with discussion, and the corresponding data are included in SI.

Fig. S9 | Reaction outcomes with TMS protected alkyne and terminal alkyne.

4) Is it possible to change the geometry of the styryl moiety?

Thank you for the suggestion. We have tried to synthesis the other isomer with different geometry of the styryl moiety, but failed with tested method. However, we could replace the styryl moiety with the cyclopropane unit, and the prepared material 1af delivered the product 2af in 91% yield. The structure of product 2af was confirmed by single-crystal X-ray diffraction analysis. These results have been added to the Table 2 in the revised manuscript, and the corresponding data have been added to the updated SI.

5) Is it possible to change the ester group to a ketone, aldehyde, or amide?

Thank you for the suggestion. We have tried to synthesis the materials as suggested. Although the preparation of the aldehyde one failed so far, we have obtained the ketone and amide derivatives 1ag and 1ah smoothly, and the reaction outcomes with these two materials are listed below.

In the reaction with ketone one, the corresponding product 2ag was isolated in 72% yield.

The amide derivative did not form the desired cyclized product, but gave the eneyne products 2ah and 2ah' in combined high yield, which formed via a Wolff rearrangement, nucleophilic addition with H₂O, decarbonation, and isomerization sequence.

Moreover, the pyrrole-fused cyclobutanone 2ae was formed in 42% yield when triazole 1ae was used as the carbene precursor under optimized conditions.

4) Do the authors have an explanation as to why 44 is a cyclic valerolactam (not a caprolactam as stated in the text, and not product 45 as also incorrectly stated in the text), whereas in the nucleophilic opening leading to 43 cyclization to the caprolactam does not occur?

Thank you for the suggestion. Corresponding mistake has been corrected. And the cyclization of 43 to the caprolactam does not occur under current conditions.

5) In the formation of benzocyclobutenones, a better description of the process, namely a Diels-Alder reaction followed by aromatization, should be included and this new methodology toward benzocyclobutenones should be compared to the ones already described in the literature.

Thank you for the suggestion. Corresponding revision has been made in the manuscript as suggested:

The furan motif, which is an electron-rich aromatic ring, could undergo a Diels-Alder reaction with electron-deficient alkenes followed by aromatization, providing a convenient access to poly-substituted benzocyclobutenones (BCBs), which would be difficult to access otherwise^{17,18,21-26}.

6) The authors should justify their chosen level of theory. Do they have benchmark data to support their choice?

Thank you for the suggestion.

In order to investigate the regiochemistry of this reaction, we have carried out a benchmark study on the cyclization steps using the state-of-the-art method DLPNO-CCSD(T).

Three elementary steps are computed using several popular functionals in the Rh-catalyzed reactions. All these steps were studied using quantum chemical calculations with the Gaussian 09 software package for DFT calculations and Orca for DLPNO-CCSD(T) calculations. For DFT calculations, pruned integration grids with 99 radial shells and 590 angular points per shell were used. Geometry optimizations of all the stationary points were carried out in the gas phase at the B3LYP/def2-SVP level. Unscaled harmonic frequency calculations at the same level were performed to validate each structure as either a minimum or a transition state. Based on the optimized structures, single-point energy refinements were performed at DLPNO-CCSD(T) and other popular functionals (BMK, M06, M06L, ω B97X-D, PBE0, B3LYP-D3(BJ), BP86) with the same basis set def2-TZVPP. The results are shown in

Table S1 below, the BMK functional, which has been used in our previous studies on the Rh-catalyzed cycloaddition, performed the best on the cyclization steps compared to other functionals though it overestimated the barrier of nitrogen-release step (M06 performs better). Thus, we choose the BMK functional to study the mechanism and regioselectivity of this reaction.

All these studies have been added to the updated SI.

Table S1 | Benchmark study

	DLPNO-CCSD(T)	BMK	M06	M06L	ω B97X-D	PBE0	B3LYP-D3	BP86
ΔE_1^\ddagger	4.7	3.5	7.0	7.6	6.4	7.5	5.7	10.2
ΔE_2^\ddagger	6.1	6.0	13.0	14.7	11.3	14.1	11.6	18.9
ΔE_3^\ddagger	15.4	22.8	15.7	14.9	17.4	18.8	15.0	12.4

7) The formation of the rhodium carbene is not computed. This is likely to be the rate determining step of the overall transformation (if not the reaction should occur at lower temperature).

Thank you for the suggestion. The formation of the rhodium carbene has been computed (**Fig. S10**). The overall activation free energy of this step *via Com1-TS* (computed at SMD(EtOAc)/BMK/def2-TZVPP//B3LYP/def2-SVP) is 28.5 kcal/mol, which should be the rate-determined step of this reaction. According to the above benchmark study, BMK functional overestimated the barrier of nitrogen-release step,

we also computed the activation free energy at SMD(EtOAc)-M06/def2-TZVPP based on the same optimized structures (at B3LYP/def2-SVP in the gas phase), the overall activation free energy of carbene formation *via* **Com1-TS** is 16.5 kcal/mol, which is also the rate-determined step of this reaction and consistent with the experiments (reaction at 40 °C).

All these results have been discussed in the revised manuscript and the corresponding data have been added to the updated SI.

Fig. S10 | The formation of the rhodium carbene. Computed at SMD(EtOAc)/BMK/def2-TZVPP//B3LYP/def2-SVP and SMD(EtOAc)/M06/def2-TZVPP//B3LYP/def2-SVP

8) In the 4-exo-dig transformations, the authors labelled the transformation from Int2 to 2a as a “Formal [3+2] cyclization”. In fact, it is the transformation from Int1 to 2a that would be a formal stepwise [3+2] cyclization. The step from Int2 to 2a is better described as an intramolecular cyclization. In addition, the authors should consider if the transformation from Int1 to Int2 can be labelled as a CAM since the carbenic character is not transferred to an alkyne carbon.

Thank you for the suggestion. Corresponding revisions have been made as suggested.

9) The authors should justify why in Figure S4 the energy for 3TS5 is lower than the energy for 3Int3.

Thank you for the suggestion. For the free energy surface in the gas phase at B3LYP/def2-SVP, the Gibbs free energy of 3TS5 is only 1.4 kcal/mol higher than that of 3Int3. Based on the optimized structures, single-point energy was carried out (at SMD(EA)-BMK/def2-TZVPP), the energy surface was slightly revised (the energy difference between 3TS5 and 3Int3 changes from 1.4 to -1.0 kcal/mol) and lead to the formal negative activation free energy. Thus, the transformation of 3Int3 to 3Int4 *via*

3T5S is not difficult based on our computation.

10) Rh₂(OAc)₄ and Rh₂(tfacam)₄ are referred to as the two real catalysts. None of them is used in the optimized reaction. A better explanation on the choice of these catalytic systems for the study of the catalyst effects on the regiochemistry should be provided.

Thank you for the suggestion. We have mentioned this in the manuscript that Rh₂(OAc)₄ was chosen because of its comparable regioselectivity in EtOAc (**2a:3a** >16:1) with the used Rh₂(OPiv)₄ (**2a:3a** >17.6:1), and with much less conformers compared to those from Rh₂(OPiv)₄, which has 12 rotated methyl group. Catalyst Rh₂(tfacam)₄ has different ligands and affords inverse selectivity (**2a:3a** = 1:3.6 in EtOAc), prompting us to investigate the reasons behind this phenomenon.

11) Whereas the calculated energies for the 4-exo-dig and 5-endo-dig cyclizations using Rh₂(OAc)₄ as a catalyst provide a fairly good explanation for the results observed, this is not the case for the Rh₂(tfacam)₄. Have the authors considered the effect that the axial coordination of a solvent on the dirhodium core might have (see for instance *Organometallics* 2021, 40, 4120–4132)? Ethyl acetate is a quite exotic solvent for metal carbene transformations that might have a non-negligible effect.

Thank you for the suggestion. We have considered about the effect of axial coordination on the cyclization steps. Our DFT calculations found that the axial coordination has insignificant effect on the regioselectivity of cyclizations (Fig. S11 and S12: $\Delta\Delta G = 0.8$ kcal/mol vs. 1.2 kcal/mol for Rh₂(OAc)₄; $\Delta\Delta G = -0.5$ kcal/mol vs. -0.1 kcal/mol for Rh₂(tfacam)₄), the changes in activation free energies of both cyclizations are also small (< 1.0 kcal/mol), thus the axial solvent might not play an important role in this case. *These results have been discussed in the revised manuscript with data updated in SI. And corresponding recent reference has been cited as ref 62:* Laconsay, C. J., Pla-Quintana, A. & Tantillo, D. J. Effects of axial solvent coordination to dirhodium complexes on the reactivity and selectivity in C–H insertion reactions: A computational study. *Organometallics* **40**, 4120–4132 (2021).

Fig. S11 | The key transition states in the regioselectivity of the real system with axial effect. Computed at SMD(EtOAc)/BMK/def2-TZVPP//B3LYP/def2-SVP.

Fig. S12 | The key transition states in the regioselectivity of the real system without axial effect. Computed at SMD(EtOAc)/BMK/def2-TZVPP//B3LYP/def2-SVP.

12) The spectra reported in the supporting information file have very low intensity for the peaks and this does not allow determining the purity of the synthesized products.

We appreciate the reviewer for the suggestion, and corresponding revision has been made in SI.

Reviewers' comments:

Reviewer #1 (Remarks to the Author):

The authors have resolved all comments and the manuscript is recommended for publication

Reviewer #3 (Remarks to the Author):

First, I thank the reviewers for the suggested experiments to probe the reaction mechanism. Also, I appreciate the authors' efforts in attempting additional experiments and carefully revising the manuscript. To the authors' credit, they have done an immense amount of work for a communication paper, which should be published as a full paper in a journal.

Unfortunately, additional control experiments provided by the authors don't support the proposed reaction pathway. Therefore, I still have reservations about the nomenclature (4-exo-dig) and the strong language used in the manuscript, such as "unprecedented" transformation.

Additionally, the Authors don't cite any paper from Albert Padwa, who pioneered carbene-initiated cycloaddition reactions. The Padwa group also published articles in 1990 on the intramolecular cycloaddition of the diazo-carbonyl to alkynes (J. Org. Chem., 1990, 55, 414–416, and J. Org. Chem. 1990, 55, 4518–4520). These papers provide further insights into the carbene-alkyne metathesis (CAM) pathway and highlight the lack of novelty in this carefully crafted starting material design to obtain achiral furan containing cyclobutanones, which don't have any significance in drug discovery, where medicinal chemists want to "escape the flatland." I still have strong reservations about supporting this paper as a communication in any journal, specifically Nature Communications. This manuscript fails to impress in all three essential review criteria of "Significance, Innovation, and Approach.

Significance: In the response letter (page 10, last paragraph) and the manuscript introduction, the authors highlight the importance of these compounds in diversity-oriented synthesis, drug discovery, and click chemistry in biorthogonal chemistry. In terms of significance, I would like to point out two things. The first problem in this scaffold is the presence of furan, which can lead to false positives due to stability issues associated with furan-containing molecules. Please check these articles (J. Med. Chem. 2015, 58, 18, 7419–7430) and a report in science

(<https://www.science.org/content/blog-post/false-positives-always-waiting>). The weak uM activity against different cancer cell lines might be due to the presence of furan.

The second issue is the absence of chiral centers in this scaffold. The authors highlight this weakness as a strength calling it the exploration of an uncharted area in synthetic chemistry. This scaffold has never been synthesized not because of the synthetic challenges but the inherent problems associated with this scaffold. No one in the drug discovery field wants to synthesize an achiral molecule having a furan as there has been an enormous push to "Escape from the Flatland" (please see the attached articles "J. Med. Chem. 2009, 52, 21, 6752–6756; Med. Chem. Commun., 2013, 4, 515-519).

Although I understand the importance of 4-membered rings, particularly chiral cyclobutanes, in synthetic chemistry and drug discovery ((ChemMedChem. 2022, 17(9): e202200020), this chemistry doesn't provide access to these valuable scaffolds.

Innovation: As stated earlier, this is a carefully crafted architect with severe limitations in terms of substrate scope. There needs to be more scope for diversification handles. The control experiments (response letter, page 15, figure S9, and page 16) show that the chemistry does not tolerate TMS-protected or free-alkyne functionality and only works well with the phenyl rings, not even another aromatic heterocycle. Also, there is no reaction with the amide functionality, and the scaffold undergoes Wolff rearrangement.

All these experiments suggest that this is simply an intramolecular formal [3+2]-cycloaddition reaction, which only works with acceptor/acceptor diazo compounds, which on carbene formation undergo fast enolization like ortho-diazo-quinones known for cycloaddition reactions. An example with triazole (Table 2, product 2ae) further supports the [3+2] cycloaddition hypothesis. Furthermore, the literature contains intramolecular cycloaddition examples of diazo compounds with alkynes and alkenes. I feel sorry, but I still would like to suggest that authors do additional experiments without the ketone functionality, which is causing the Wolff rearrangements issue. This will change the diazo to acceptor only and provide further insights into the underlying reaction pathway. Also, I would like to see what happens when an alkene functionality replaces the alkyne functionality.

Overall, the lack of innovation in methodology and synthetic value of the furan-contacting product further diminishes my enthusiasm.

Approach: The major problem in the manuscript is the reliance on computational calculations by ignoring the experimental results. No explanation is provided in the response letter or in the manuscript why the change from an ester to amide alters the reaction outcome. Why does the reaction not work with the TMS-protected or free alkyne? What happened with the heteroatom-substituted (O/N) alkynes, which can further stabilize the cationic intermediates? Furthermore, the trapping experiments with 20 equivalents of styrene do not lead to the insertion of styrene into the

products, contrary to the Author's initial work (Nat. Commun 12, 1182 (2021). <https://doi.org/10.1038/s41467-021-21335-9>), where external alkenes trap the resulting gold-carbene intermediate.

With due respect and no offense to the computational colleagues in the field, we all know that calculations are not very reliable in elucidating the reaction mechanism. The experiment with the benzyl alcohol provides an OH insertion product, further supporting the CAM pathway. To probe the cation or a carbene, authors can attempt a reaction with an alkyne with a 2-allyl-phenyl ring, leading to cyclopropane formation.

Finally, I can't entirely agree with the last statement in this communications paper (page 17, lines 368–370) "Further synthetic applications, including but not limited to novel transformations development and bioactive molecule discovery, could be expected in due course." In my opinion, the Authors have already performed everything. Not much can follow this communication paper as furan is a big obstacle in achieving bioactive molecules from this scaffold. Therefore, in its present form, this manuscript is more suitable as a full paper in any journal.

With all these considerations above and the lack of novelty and significance of this work, I do not support this manuscript as a communication for publication in any journal, including Nature Communications.

Reviewer #4 (Remarks to the Author):

The revised version of the manuscript entitled "Catalytic 4-exo-dig carbocyclization for the construction of furan-fused cyclobutanones and synthetic applications" by Zhi-Xiang Yu, Xinfang Xu, and co-workers addresses in a satisfactory manner the concerns raised in the initial reviewer report. The present version clearly sets out the scope of the reaction and gives a clearer view of the mechanism. I only suggest addressing the minor points (better graphical representation of the results, expressions that can be improved and typographical errors to be corrected) listed below:

- Fig. 7: The energetics for the carbene formation (1a to Int 1) should be included in the graphical representation.
- Line 147: It is not clear with the provided text and the structures in Table 2, what “corresponding triazole” means. Please add a sentence to clarify the structure of the starting material used.
- Line 149: It is not clear what does “note b” refer to.
- Line 172: please correct “unite” to “unit”
- Lines 183-185: please revise the whole sentence.
- Line 226: please correct “confirmation by” to “confirmed by”.
- Lines 272-273: please revise the sentence.
- Line 293: please correct “Fig. 6” to “Fig. 7”

For Reviewer #3 (Remarks to the Author):

1) First, I thank the reviewers for the suggested experiments to probe the reaction mechanism. Also, I appreciate the authors' efforts in attempting additional experiments and carefully revising the manuscript. To the authors' credit, they have done an immense amount of work for a communication paper, which should be published as a full paper in a journal.

Unfortunately, additional control experiments provided by the authors don't support the proposed reaction pathway. Therefore, I still have reservations about the nomenclature (4-exo-dig) and the strong language used in the manuscript, such as "unprecedented" transformation.

Additionally, the Authors don't cite any paper from Albert Padwa, who pioneered carbene-initiated cycloaddition reactions. The Padwa group also published articles in 1990 on the intramolecular cycloaddition of the diazo-carbonyl to alkynes (J. Org. Chem., 1990, 55, 414–416, and J. Org. Chem. 1990, 55, 4518–4520). These papers provide further insights into the carbene-alkyne metathesis (CAM) pathway and highlight the lack of novelty in this carefully crafted starting material design to obtain achiral furan containing cyclobutanones, which don't have any significance in drug discovery, where medicinal chemists want to "escape the flatland." I still have strong reservations about supporting this paper as a communication in any journal, specifically Nature Communications. This manuscript fails to impress in all three essential review criteria of "Significance, Innovation, and Approach.

We appreciate the reviewer for the suggestions and comments.

According to the suggestion related to the nomenclature, we have revised the manuscript as “a Rh-catalyzed formal [3+2] annulation of diazo group tethered alkynes involving a 4-exo-dig carbocyclization process”.

For the references related to the works of Padwa (J. Org. Chem., 1990, 55, 414–416) and Hoye (J. Org. Chem. 1990, 55, 4518), as mentioned by this reviewer, they are both early examples of on the intramolecular cycloaddition of the diazo-carbonyl to alkynes. We have proposed the term carbene-alkyne metathesis (CAM) in our review article (ref. 38), and these two examples have been included in this review paper. During the past decades, a plenty of advances have been disclosed in this area by chemists around the world (for reviews: ref 37 & 38; for selected advances: ref 39-52). However, these cascade reactions mainly focused on the preparation of 5- and 6-membered carbocyclic structures through different types of exo- or endo-dig

cyclizations (Fig. 1b in the manuscript), and no analogous process involving 4-exo-dig carbocyclization process has been reported yet.

We are sorry that we could not include all these references, and we have added a note “and the references cited herein.” to the cited recent review in this area.

2) Significance: In the response letter (page 10, last paragraph) and the manuscript introduction, the authors highlight the importance of these compounds in diversity-oriented synthesis, drug discovery, and click chemistry in biorthogonal chemistry. **In terms of significance, I would like to point out two things. The first problem in this scaffold is the presence of furan, which can lead to false positives due to stability issues associated with furan-containing molecules.** Please check these articles (J. Med. Chem. 2015, 58, 18, 7419–7430) and a report in science (<https://www.science.org/content/blog-post/false-positives-always-waiting>). The weak μM activity against different cancer cell lines might be due to the presence of furan.

We appreciate the reviewer for the comments. We do not argue the results presented in the literature (J. Med. Chem. 2015, 58, 18, 7419–7430), however, this accidental example(s) could not lead to the conclusion to the current work that “The first problem in this scaffold is the presence of furan, which can lead to false positives due to stability issues associated with furan-containing molecules.”, which is totally ignored the general scientific facts and experimental results.

Firstly, the importance of furan motif in medical chemistry is undoubted, although someone might not like it, but it is not the rational reason to question the significance of this method for the direct construction of unique furan-fused cyclobutanones.

Secondly, these synthesized compounds are stable and could be storage for few weeks without any obvious decomposition. The bioactivity test of these synthesized furan derivatives against cancer cell lines is differential (see Table S1 and S2 in SI for details). These experimental results against the reviewer’s comment “which can lead to false positives due to stability issues associated with furan-containing molecules”, otherwise, this “false positives” phenomenon would be the same for all these tested analogues.

In addition, we have synthesized another amide 72 derived from 3-furanoic acid that did not show any anti-cancer cell activity (29 vs 72), which implied the unique bioactivity of these poly-substituted furan derivatives might not due to the stability issues associated with these molecules.

29

HCT-116:	$IC_{50} = 1.19 \pm 0.06 \mu\text{M}$
MCF-7:	$IC_{50} = 10.4 \pm 0.78 \mu\text{M}$
A549:	$IC_{50} = 2.74 \pm 0.09 \mu\text{M}$
SJSA-1:	$IC_{50} = 2.08 \pm 0.09 \mu\text{M}$

72

$IC_{50} = > 50 \mu\text{M}$
$IC_{50} = > 50 \mu\text{M}$
$IC_{50} = > 50 \mu\text{M}$
$IC_{50} = > 50 \mu\text{M}$

3) The second issue is the absence of chiral centers in this scaffold. The authors highlight this weakness as a strength calling it the exploration of an uncharted area in synthetic chemistry. **This scaffold has never been synthesized not because of the synthetic challenges but the inherent problems associated with this scaffold. No one in the drug discovery field wants to synthesize an achiral molecule having a furan as there has been an enormous push to "Escape from the Flatland"** (please see the attached articles "J. Med. Chem. 2009, 52, 21, 6752 – 6756; Med. Chem. Commun., 2013, 4, 515-519).

Although I understand the importance of 4-membered rings, particularly chiral cyclobutanes, in synthetic chemistry and drug discovery ((ChemMedChem. 2022, 17(9): e202200020), this chemistry doesn't provide access to these valuable scaffolds.

We appreciate the reviewer for the comments, but it is quite difficult to follow this reviewer's logic, since these references did not show any pertinence or direct argument against the significance of this work

Firstly, for "This scaffold has never been synthesized not because of the synthetic challenges but the inherent problems associated with this scaffold. No one in the drug discovery field wants to synthesize an achiral molecule having a furan as there has been an enormous push to "Escape from the Flatland" ", it hard to understand how could the reviewer led to this conclusion by just quoting two unrelated references. Although there might have stability and so called "Escape from the Flatland" issues related to some furan derivatives in medical chemistry, which could not neglect the pervasive existence of furan motif in drug discovery, there are so many reported bioactive molecules containing furan motif, and below listed are some of the selected examples of achiral furan derivatives.

Thus, it is obvious that this reviewer might have some prejudice with the furan molecules based on few accidental examples, however, this is not the right way for a reviewer by ignoring the general scientific facts and experimental results, and questing a synthetic method for the expeditious assembly of previously inaccessible and unknown furan-fused cyclobutanones.

Nifuroxazide is an antibiotic indicated in the treatment of susceptible gastrointestinal infections. (ref: *Bioorgan. Med. Chem.* **2008**, *16*, 6724-6731; *Toxicol. Appl. Pharm.* **2020**, *401*, 115104)

Nifuratel is a nitrofurane derivative indicated in the treatment of leucorrhoea, vulvovaginal infections, and urinary tract infections. (ref: *Pharmaceuticals* **2022**, *15*, 705)

Nitrofurantoin is an antibiotic used to treat urinary tract infections. (ref: *J. Med. Chem.* **2023**, *66*, 4565-4587)

Nifurtimox is an antiparasitic drug used for the treatment of Chagas disease (*Trypanosoma cruzi* infection). (ref: *J. Pharmacol. Exp. Ther.* **2011**, *336*, 506-615)

Ranitidine is a histamine H₂ antagonist used to treat duodenal ulcers, Zollinger-Ellison syndrome, gastric ulcers, GERD, and erosive esophagitis. (ref: *Drugs* **1989**, *37*, 801-870; *Pharm. World Sci.* **2005**, *27*, 432-435)

Navarixin is a potent, allosteric and orally active antagonist of both CXCR1 and CXCR2, with K_d values of 41 nM for cynomolgus CXCR1 and 0.20 nM, 0.20 nM, 0.08 nM for mouse, rat and cynomolgus monkey CXCR2, respectively. (ref: *J. Med. Chem.* **2006**, *49*, 7603-6706)

4) Innovation: As stated earlier, this is a carefully crafted architect with severe limitations in terms of substrate scope. There needs to be more scope for diversification

handles. The control experiments (response letter, page 15, figure S9, and page 16) show that **the chemistry does not tolerate TMS-protected or free-alkyne functionality and only works well with the phenyl rings, not even another aromatic heterocycle**. Also, there is no reaction with the amide functionality, and the scaffold undergoes Wolff rearrangement.

All these experiments suggest that this is simply an intramolecular formal [3+2]-cycloaddition reaction, which only works with acceptor/acceptor diazo compounds, which on carbene formation undergo fast enolization like ortho-diazo-quinones known for cycloaddition reactions. An example with triazole (Table 2, product 2ae) further supports the [3+2] cycloaddition hypothesis. Furthermore, the literature contains intramolecular cycloaddition examples of diazo compounds with alkynes and alkenes. I feel sorry, but I still would like to suggest that authors do additional experiments without the ketone functionality, which is causing the Wolff rearrangements issue. This will change the diazo to acceptor only and provide further insights into the underlying reaction pathway. Also, I would like to see what happens when an alkene functionality replaces the alkyne functionality.

Overall, the lack of innovation in methodology and synthetic value of the furan-contacting product further diminishes my enthusiasm.

We appreciate the reviewer for the comments, but it is difficult to understand that this reviewer wants to discuss about the “innovation” or “substrate scope” of this work.

For the innovation, this work presented a Rh-catalyzed 4-exo-dig carbocyclization process, using diazo tethered alkynes as substrates, which provides straightforward access to the synthesis of structurally novel furan-fused cyclobutanone scaffolds that are nontrivial to construct otherwise. This is the first example of catalytic 4-exo-dig carbocyclization of alkynes for the construction of cyclobutanone structures.

This method has proved successful to direct synthesis of previously inaccessible and unknown furan-fused cyclobutanone scaffolds, which could be served as a platform molecule, participating in different types of ring-opening and cycloaddition reactions as versatile synthetic building blocks for diversity-oriented-synthesis (by referencing the analogous luxuriant chemistry of benzocyclobutenone (BCB) skeletons and our unpublished results listed below). A variety of drugs and sterols are tolerated in the ring-opening transformation with these synthesized furan-fused cyclobutanones under mild conditions, demonstrating the innovation in methodology and synthetic value of this method.

For the substrate scope, it is well known that the substitution(s) adjacent to the carbene center played a pivotal role in the reactivity and selectivity of carbene chemistry (Chem. Rev. 2003, 103, 2861; Chem. Soc. Rev. 2011, 40, 1857; Chem. Soc. Rev. 2020, 49, 908), that's why the substitution effect is obvious in metal carbene reaction. In this reaction, diazo ketones containing a variety of functional groups, including different substituted aryl, naphthyl, thienyl, ferrocenyl, cyclohexenyl, and alkyl halide, all worked smoothly (which are not as this reviewer said: the chemistry does not tolerate TMS-protected or free-alkyne functionality and only works well with the phenyl rings, not even another aromatic heterocycle.), delivering the corresponding products in good to high yields, which showed broad functional group compatibility and generality of current method.

5) Approach: The major problem in the manuscript is the reliance on computational calculations by ignoring the experimental results. No explanation is provided in the response letter or in the manuscript why the change from an ester to amide alters the reaction outcome. Why does the reaction not work with the TMS-protected or free alkyne? What happened with the heteroatom-substituted (O/N) alkynes, which can further stabilize the cationic intermediates? Furthermore, the trapping experiments with 20 equivalents of styrene do not lead to the insertion of styrene into the products, contrary to the Author's initial work (Nat. Commun 12, 1182 (2021). <https://doi.org/10.1038/s41467-021-21335-9>), where external alkenes trap the resulting gold-carbene intermediate.

With due respect and no offense to the computational colleagues in the field, we all know that calculations are not very reliable in elucidating the reaction mechanism. The experiment with the benzyl alcohol provides an OH insertion product, further supporting the CAM pathway. To probe the cation or a carbene, authors can attempt a reaction with an alkyne with a 2-allyl-phenyl ring, leading to cyclopropane formation.

We appreciate the reviewer for the comments, but there must be a misunderstanding between the reviewer and us.

To probe the cation or a carbene, researchers usually adopt the cyclopropanation reaction with alkenes to verify the carbene intermediates; whereas, for the OH insertion products, both the cation or a carbene could led to the same products.

In our previous work (Nat. Commun 12, 1182 (2021). <https://doi.org/10.1038/s41467-021-21335-9>), the in-situ formed gold carbene intermediate show cation reactivity

due to the inherent carbene structure and the ligand effect on the gold-complex, which is in consistent with previous reports (J. Am. Chem. Soc. 2014, 136, 6904; Angew. Chem. Int. Ed. 2014, 53, 9817). That's why this intermediate addition with alkene went through a stepwise pathway, rather than form the cyclopropanation product, leading to a formal [4+2]-cycloaddition reaction.

Moreover, the cation species in this stepwise process was successfully intercepted by an external alcohol to give the three-component product in this reaction. All these results in precious studies are consistent with the experimental results in this manuscript and our proposed reaction pathway. It would be impossible for anyone led to the conclusion like that “The major problem in the manuscript is the reliance on computational calculations by ignoring the experimental results.”

6) Finally, I can't entirely agree with the last statement in this communications paper (page 17, lines 368–370) "Further synthetic applications, including but not limited to novel transformations development and bioactive molecule discovery, could be expected in due course." **In my opinion, the Authors have already performed everything. Not much can follow this communication paper as furan is a big obstacle in achieving bioactive molecules from this scaffold.** Therefore, in its present form, this manuscript is more suitable as a full paper in any journal.

We appreciate the reviewer for the comments, but we totally could not understand how or what that led this reviewer to the opinion “In my opinion, the Authors have already performed everything. Not much can follow this communication paper as furan is a big obstacle in achieving bioactive molecules from this scaffold.”.

As we mentioned during the transfer from Nature Chemistry to Nature Communications, the obtained unknown furan-fused cyclobutanone scaffolds could be served as a platform molecule and versatile synthetic building block, participating in different types of ring-opening and cycloaddition reactions for diversity-oriented-synthesis.

During this submission, we are continuing working on the exploration of novel catalytic transformations with this intriguing furan-fused cyclobutanone as a 4C synthon in a variety of cycloadditions, including [4+2], [4+3], [4+4], etc. (see the listed results below). And we would like to report these results in due course by focusing on the synthetic applications and asymmetric catalytic transformations.

Reviewer #1 (Remarks to the Author):

The authors have resolved all comments and the manuscript is recommended for publication

We appreciate the reviewer for the suggestions and comments.

Reviewer #4 (Remarks to the Author):

The revised version of the manuscript entitled "Catalytic 4-exo-dig carbocyclization for the construction of furan-fused cyclobutanones and synthetic applications" by Zhi-Xiang Yu, Xinfang Xu, and co-workers addresses in a satisfactory manner the concerns raised in the initial reviewer report. The present version clearly sets out the scope of the reaction and gives a clearer view of the mechanism. I only suggest addressing the minor points (better graphical representation of the results, expressions that can be improved and typographical errors to be corrected) listed below:

- Fig. 7: The energetics for the carbene formation (1a to Int 1) should be included in the graphical representation.

- Line 147: It is not clear with the provided text and the structures in Table 2, what "corresponding triazole" means. Please add a sentence to clarify the structure of the starting material used.

- Line 149: It is not clear what does “note b” refer to.
- Line 172: please correct “unite” to “unit”
- Lines 183-185: please revise the whole sentence.
- Line 226: please correct “confirmation by” to “confirmed by”.
- Lines 272-273: please revise the sentence.
- Line 293: please correct “Fig. 6” to “Fig. 7”

We appreciate the reviewer for the suggestions and comments. All the revisions have been made as suggested in the revised manuscripts.